# Global modelling studies of composition and decadal trends of the Asian Tropopause Aerosol Layer

Adriana Bossolasco[1], Fabrice Jegou[1], Pasquale Sellitto[2], Gwenaël Berthet[1], Corinna Kloss[1], and Bernard Legras[3]

[1]Laboratoire de Physique et Chimie de l'Environnement et de l'Espace, CNRS/Université d'Orléans, UMR 7328, Orléans, France
[2]Laboratoire Interuniversitaire des Systèmes Atmosphériques, UMR CNRS 7583, IPSL, Université Paris-Est Créteil/Université de Paris, Créteil, France
[3]Laboratoire de Météorologie Dynamique, UMR CNRS 8539, IPSL, ENS-PSL/Sorbonne Université//École Polytechnique, Paris, France

*Correspondence to: Adriana Bossolasco (adriana.bossolasco@cnrs-orleans.fr)*

**Abstract.** The Asian Summer Monsoon (ASM) traps convectively-lifted boundary layer pollutants inside its upper-tropospheric lower-stratospheric Asian monsoon anticyclone (AMA). It is associated with a seasonal and spatially-confined enhanced aerosol layer, called the Asian Tropopause Aerosol Layer (ATAL). Due to the dynamical variability of the AMA, the dearth of in situ observations in this region, the complexity of the emission sources and of transport pathways, the knowledge of the ATAL properties in terms of aerosol budget, chemical composition, as well as its variability and temporal trend, is still largely uncertain. In this work, we use the Community Earth System Model (CESM 1.2 version) based on the coupling of the Community Atmosphere Model (CAM5) and the MAM7 (Modal Aerosol Model) aerosol module to simulate the composition of the ATAL and its decadal trends. Our simulations cover a long-term period of 16 years from 2000 to 2015. We identify a typical "double-peak" vertical profile of aerosols for the ATAL. We attribute the upper peak (around 100 hPa, predominant during early ATAL, e.g. in June) to dry aerosols, possibly from nucleation processes, and the lower peak (around 250 hPa, predominant for a well-developed and late ATAL, e.g. in July and August) to cloud-borne aerosols associated with convective clouds. We find that mineral dust (present in both peaks) is the dominant aerosol by mass in the ATAL, showing a large interannual variability but no long-term trend, due to its natural variability. The results between 120-80 hPa (dry aerosol peak) suggest that for aerosols other than dust the ATAL is composed of around 40 % of sulfate, 30% of secondary and 15% of primary organic aerosols, 14% of ammonium aerosols and less than 3% of black carbon. Nitrate aerosols are not considered in MAM7. The analysis of the anthropogenic and biomass burning aerosols shows a positive trend for all aerosols simulated by CESM-MAM7.

## 1-Introduction

During boreal summer, major convective activity is driven by the Asian summer monsoon (ASM). The ASM-related convection combines both land convection over mainland Asia and maritime convection over surrounding seas. This dynamical mechanism acts as a pathway for the transport of trace gases and pollutants from the boundary layer to the UTLS (Upper

Troposphere Lower Stratosphere) (Randel and Park, 2006; Park et al., 2007; Pan et al., 2016; Gottschaldt et al., 2017). The upper atmospheric circulation is dominated by the related Asian Monsoon Anticyclone (AMA), which is known to contain enhanced concentration of tropospheric trace gases and aerosols (Randel and Park, 2006, Park et al., 2007; Park et al., 2008), due to rapid lifting from the boundary layer by deep convection and subsequent horizontal

confinement. The AMA is confined by the subtropical westerly jet stream in the north (~40–45°N) and the equatorial easterly jet stream in the south (~10–15°N), and spans from about 20–140 °E in the northern hemisphere. The altitude of maximum strength of the anticyclonic circulation is around the local tropopause (17–18 km)  (e.g., Dethof et al., 1999; Bian et al., 2012; Ploeger et al., 2015; Garny and Randel, 2016; Pan et al., 2016, Brunamonti et al., 2018).

On a daily basis, the specific location, spatial extent and strength of the AMA depend on the internal dynamical variability of the ASM (Randel and Park, 2006; Garny and Randel, 2013; Vogel et al., 2015; Pan et al., 2016). As suggested, the AMA can effectively trap boundary layer pollutants and is associated with the formation of the Asian Tropopause Aerosol Layer (ATAL) (Vernier et al., 2011, Vernier et al., 2015). The ATAL refers to an enhanced aerosol layer near

the tropopause over the Asian monsoon region extending from ~13 to ~18 km altitudes. Its horizontal extension is determined by the AMA geometry, roughly in the broad region bounded by approximately 5-105° E, 15-45° N (e.g. Vernier et al., 2015, Lau et al., 2018, Bian et al., 2020). Combined satellite observations from SAGE (Stratospheric Aerosol and Gas Experiment) II and CALIOP (Cloud-Aerosol LIDAR with Orthogonal Polarization) have highlighted the

presence of the ATAL since 1998 (Vernier et al., 2015). Höpfner et al. (2019) have revealed the presence of ammonium nitrate aerosols inside the AMA in August 1997 from CRISTA (Cryogenic Infrared Spectrometers and Telescopes for the Atmosphere) satellite observations. Model studies have suggested that the ATAL might have been present previously but was masked by the overwhelming UTLS aerosols produced by the Mount Pinatubo eruption (Neely et al., 2014).

The sources, chemical composition and spatial and temporal variability of the ATAL are not yet well understood. Recent observations from the StratoClim (Stratospheric and upper tropospheric processes for better climate predictions) aircraft campaign in 2017 and a few recent balloon measurements from the BATAL (Balloon measurement campaigns of the Asian Tropopause Aerosol Layer) 2015 campaign, suggest that aerosol particles in the ATAL may

contain large amounts of sulfate, as well as organics, nitrates (including ammonium nitrate), black carbon and dust (Vernier et al., 2015, 2018; Höpfner et al., 2019). Different indications on the ATAL composition have been brought by a number of modelling studies. Fadnavis et al. (2013), using the aerosol–chemistry–climate model ECHAM5-HAMMOZ, studied the transport of aerosols to the UTLS and showed persistent maxima in black carbon, organic carbon, sulfate,

and mineral dust aerosols within the anticyclone throughout the ASM (from July to September). Yu et al. (2015), using the CESM1 (Community Earth System Model) global Earth system model coupled with the CARMA (Community Aerosol and Radiation Model for Atmospheres) aerosol model, have suggested that the ATAL might be principally composed of secondary organic and sulfate aerosols, as well as of primary organic aerosols. Fadnavis et al. (2017) performed model

simulations with ECHAM6-HAM (European Centre Hamburg Model 6.3-Hamburg Aerosol Model)

global aerosol-climate model, and their simulations showed a persistent maximum of carbonaceous aerosols in the ATAL region. Ma et al. (2019), using the ECHAM/MESSy (Modular Earth Submodel System) for Atmospheric Chemistry (EMAC) general circulation model coupled with the Global Modal-aerosol eXtension (GMXe) aerosol module, have found that mineral dust

and water-soluble compounds, like nitrate and sulfate, are the principal typology of aerosols over the Tibetan Plateau, within the AMA. Using the GEOS-Chem (Goddard Earth Observing System with Chemistry) chemical transport model, Fairlie et al. (2020) have found significant amounts of sulfate, ammonium, organic aerosols and nitrate in the ATAL, with a predominant contribution of nitrate, as was identified previously by Gu et al. (2016) using an earlier version

of the model. Therefore, existing modelling studies have proved to be able to simulate the enhanced concentration of aerosols in the AMA region, even if a very large uncertainty in the composition of the ATAL remains.

In several studies, dust has been shown as a major contributor to the aerosol burden in the Asian upper troposphere during summer.  Xu et al. (2015), using CALIOP and MISR (Multi-angle

Imaging SpectroRadiometer) satellite data, have found that dust is one of the predominant aerosol over the Tibetan Plateau most probably originating from the Taklamakan desert and lofted from the surface to an altitude of about 10 km. Ma et al. (2019) have simulated a broad maximum of dust surface concentration at the northern edge of the Tibetan Plateau up to 10 km. Their model results have shown that the enhancement of dust aerosols is still visible up to

16 km above the Tibetan Plateau, with maximum shifted to the east and south as a consequence of the influence of anticyclonic circulation. Large amounts of dust have been also reported by Lau et al. (2018) in the mid- and upper- troposphere over India and China from May to June transported from the Middle East desert, and then from July to August trapped and accumulated within the AMA and contributing to the ATAL formation.

A rising temporal trend of the ATAL optical signature in the AMA region has been observed (Vernier et al., 2015). The recent rising trends of sulfur dioxide and volatile organic compounds emissions in India have been proposed as a candidate for explaining the appearance of the ATAL and its evolution. Continental convective regions have also been shown to be the main contributors to the air trapped within the AMA with North India and South of the Tibetan

Plateau as specific source areas (e.g. Tissier and Legras, 2016; Legras et al., 2019). Bergman et al. (2013), using Lagrangian backward trajectories, have shown that the anticyclone is connected to the boundary layer through a vertical conduit centred over Northeast India, Nepal, and southern Tibet. In the recent BATAL campaign, Vernier et al. (2018) have used back-trajectory calculations to point at North of India as a principal region source for ATAL. Lau et al.

(2018), based on MERRA-2 reanalysis have reported that the Himalayas Gangetic Plain (HGP) region and the Sichuan Basin (SB) of southwestern China, are two important regions with strong vertical transport of CO, carbonaceous aerosols and dust from the surface to the UTLS. On the other hand, the simulations of Fairlie et al. (2020)  have suggested that the anthropogenic sources from India contribute to up to 40% of sulfate and up to 65% of organic

and ammonium aerosols in the western ATAL region, whereas China contributes up to 60% (both sulfate and organic aerosols) in the eastern ATAL region.

It's also important to note that the ATAL formation and possible spatial and temporal variability is closely related to the dynamical variability of the AMA. For example, Basha et al. (2019) have suggested that the spatial extent and strength of the AMA is greater during July and August compared to June and September, and that the decadal variability is bigger at the edges of the anticyclone. As a consequence of the variability of atmospheric dynamics, some years show a stronger monsoon activity than others (Lau et al., 2018, Basha et al., 2019, Yuan et al., 2019) and this affects the ATAL formation, location and composition. Several studies have shown that the AMA exhibits intraseasonal variability between the Iranian Plateau and the Tibetan Plateau with a quasi-biweekly oscillation (e.g. Zhang et al., 2002; Yan et al., 2011; Nützel et al., 2016; Pan et al. 2016; Wei et al., 2019).

This study provides further insight on the chemical composition of the ATAL and assesses its decadal variability composition and aerosol trends for the first time. To asses this, we have carried out long-term modelling of the ATAL using the Community Earth System Model (CESM 1.2) which embeds the Community Atmosphere Model (CAM5) coupled with the MAM7 (Modal Aerosol Model) aerosol module. Our simulations cover an overall extended period of 16 years, from January 15th 2000 to December 15th 2015. Yuan et al., 2019 derived decadal trends for carbonaceous aerosols and dust in the ATAL using only meteorological reanalysis data, while in the present study a detailed chemistry and microphysical modelling is used to estimate trends for a more comprehensive set of aerosol compositions.

The present paper is structured as follows. In Sect. 2, we describe the model and correlative data used for its validation. The validation is discussed in Sect.3. Results are presented and discussed in Sect. 4. Conclusions are drawn in Sect. 5.

**2-Model set-up and satellite observations**

**2.1-The CESM-MAM7 model**

Model simulations were performed using the global Community Earth System Model (CESM1.2), based on the Community Atmospheric Model (CAMS 5.1) with its full chemical core for both troposphere and stratosphere, coupled with the Modal Aerosol Model (MAM7). The MAM7 module treats the aerosol microphysics, size distribution and both internal and external mixing using seven modes. The seven modes are, specifically: Accumulation (a1), Aitken (a2), Primary Carbon (a3), Fine Dust and Sea Salt (a5 and a4), and Coarse Dust and Sea Salt (a7 and a6) (Liu et. al 2012). Extraterrestrial aerosols are neglected in our model. Table 1 lists the aerosols and dry diameter size ranges of each mode. The size distributions of each mode are assumed to be log-normal.

| Mode | Accumulation (a1) | Aitken (a2) | Primary Carbon (a3) | Fine Sea Salt (a4) | Fine Soil Dust (a5) | Coarse Sea Salt (a6) | Coarse Soil Dust (a7) |
|---|---|---|---|---|---|---|---|
| **Aerosols species** | Sulfate ($SO_4$) <br><br> Ammonium ($NH_4$) <br><br> Secondary Organic Aerosols (SOA) <br><br> Primary Organic Aerosols (POM) <br><br> Black Carbon (BC) <br><br> Sea Salt | Sulfate ($SO_4$) <br><br> Ammonium ($NH_4$) <br><br> Secondary Organic Aerosols (SOA) <br><br> Sea Salt | Primary Organic Aerosols (POM) <br><br> Black Carbon (BC) | Seal Salt <br><br> Sulfate ($SO_4$) <br><br> Ammonium ($NH_4$) | Soil Dust <br><br> Sulfate ($SO_4$) <br><br> Ammonium ($NH_4$) | Seal Salt <br><br> Sulfate ($SO_4$) <br><br> Ammonium ($NH_4$) | Soil Dust <br><br> Sulfate ($SO_4$) <br><br> Ammonium ($NH_4$) |
| **Size range (µm)** | 0.056–0.26 | 0.015–0.052 | 0.039–0.13 | 0.095–0.56 | 0.14–0.62 | 0.63–3.70 | 0.59–2.75 |

160

**Table 1: Predicted species for interstitial and cloud-borne components (see text) of each aerosol mode in MAM7 and dry diameter size ranges.**

The total number of transported aerosol tracers by the 7 log-normal modes in MAM7 is 31. The transported precursor gas species are $SO_2$ (sulfur dioxide), $H_2O_2$ (hydrogen peroxide), DMS (dimethyl sulfide), $H_2SO_4$ (sulfuric acid gas vapour), $NH_3$ (ammonia) and lumped semi-volatile organic species (Big Alkenes, Big Alkanes, Toluene, Isoprene and Monoterpenes).

Wet removal of soluble gas-phase species combines two processes: in-cloud, or nucleation scavenging (rainout), which is the local uptake of soluble gases and aerosols by the formation of initial cloud droplets and their conversion to precipitation, and below-cloud, or impaction scavenging (washout), which is the collection of soluble species from the interstitial air by falling droplets or from the liquid phase via accretion processes. The transfer of soluble gases into liquid condensate is calculated using Henry's Law, assuming equilibrium between the gas and liquid phase. This is the standard scheme used in CAM5.1 (Lamarque et al., 2012), although as noted by Fairlie et al. (2020) a more physically-based treatment of wet scavenging of $SO_2$ in convective updrafts increases the amount of sulfate.

The MAM7 module explicitly treats the microphysics of sulfate ($SO_4$), ammonium ($NH_4$), sea-salt, dust, black carbon (BC), primary organic matter (POM), and secondary organic aerosol (SOA). It simulates nucleation, condensation, coagulation, dry deposition, wet removal, and water uptake of aerosols. The formation of new particles by nucleation occurs in the Aiken mode, which is calculated using a ternary parameterization ($H_2SO_4$-$NH_3$-$H_2O$) and boundary nucleation (Merikanto et al., 2007). The inter- and intra-modal coagulation is calculated for Aitken, Accumulation and Primary Carbon modes.

In MAM7 the aerosol particles (AP) can exist in the "interstitial" state (AP that are suspended in clear or cloudy air) and "cloud-borne" state (AP attached to or contained within different

hydrometeors, such as cloud droplets and/or ice crystals). MAM7 distinguishes between cloud-borne aerosols that are within stratiform clouds, and the interstitial aerosols which include both clear-sky AP and AP contained within convective clouds. This means that the AP in convective cloud droplets are lumped with the interstitial AP in the model and the interstitial aerosol mixing ratios include the truly interstitial (i.e. "clear-sky/dry") AP and the "convective" cloud-borne AP.

As has been detailed in Wang et al. (2013), in CAM5-MAM7 cloud-borne aerosols in stratiform clouds are treated in a prognostic way in CAM5: their mixing ratios are saved between model time steps and evolve as a result of source, sink, and transport processes. Their activation is parametrised using vertical velocity (resolved and sub-grid turbulent) and aerosol properties of all the modes, following Abdul-Razzak and Ghan (2000). The stratiform-cloud-borne AP are assumed to not interact with convective clouds. AP in convective clouds are treated diagnostically: their mixing ratios are diagnosed each model time step (with no "memory") from the interstitial aerosol mixing ratios.

Both interstitial and cloud-borne aerosol particles are subject to wet and dry removal deposition. CESM-MAM7 distinguishes between "in-cloud" and "below-cloud" wet removal. In-cloud wet removal involves activation of interstitial aerosol to become cloud-borne, followed by conversion of cloud droplets (and the cloud-borne aerosol particles) to precipitation. Below-cloud wet removal involves direct capture of interstitial aerosols by precipitation particles through a number of processes (e.g., inertial impaction, Brownian diffusion) and is relatively inefficient for aerosol in the accumulation mode size range. For a complete description of the CAM5-MAM7 model see Liu et al. (2012).

In our configuration, land, sea-ice, and rivers are interactive processes in CESM, whereas oceans are prescribed. The model horizontal grid resolution is 1.9°x 2.5° in latitude x longitude and is has 56 vertical levels of altitude extending from the surface to approximately 45 km altitude, with 30 levels in the troposphere and 10 levels in the UTLS, at a vertical resolution of approximately 1 km.

The following emissions are used in our simulations. Biogenic emissions for CO, isoprene, $C_2H_4$, $C_2H_6$, $C_3H_6$, $C_3H_8$, acetone, methanol and isoprene are taken from MEGAN-MACC emission inventory (Sindelarova et al., 2014). Anthropogenic emissions and biomass burning emissions are based on the CMIP6 (Coupled Model Intercomparison Project Phase 6) inventories provided by the Community Emissions Data System (CEDS, http://www.globalchange.umd.edu/ceds/ceds-cmip6-data/). According to CEDS, the anthropogenic emissions are first scaled to EDGAR database for most emission species, then to national/regional inventories. For instance, REAS 2.1 (Regional Emission inventory in ASia version 2.1, Kurokawa et al., 2013) is the national inventory used in Asia, for $SO_2$, NOx, NMVOCs, CO and $CH_4$. For each inventory, scaling factors are calculated for years when inventory data are available. Where inventory data are not available over the specified scaling time frame, remaining scaling factors are interpolated and extended to provide a continuous trend (Hoesly et al., 2018). The goal of the scaling process is to match CEDS emission estimates with comparable inventories. The scaling process modifies CEDS default emissions and emission factors, possibly leading to an additional source of uncertainties.

The biomass burning emissions for CMIP6 are based on merged satellite observation and fires models (van Marle et al., 2017), using GFED4 (Global Fire Emissions Database version 4), which include small-magnitude fires (available from 1997 to 2015).

The emission of sea salt aerosols from the ocean follows the parameterization of Märtensson et al. (2003), for aerosols with geometric diameter < 2.8 µm. For aerosols with a geometric diameter ≥ 2.8 µm, sea salt emissions follow the parameterization of Monahan et al. (1986). The emission of mineral dust particles is calculated based on the Dust Entrainment and Deposition Model (Zender et al., 2003). Volcanic $SO_2$ emissions were obtained through the Volcanic Emissions for Earth System Models (VolcanEESM) initiative, described by Mills et al. (2016). The VolcanEESM database contains estimates of total $SO_2$ emissions by volcanic eruptions over the 1850-2016 period.

The meteorology in the model has been nudged using MERRA2 (Modern-Era Retrospective analysis for Research and Applications, Version 2, https://rda.ucar.edu/datasets/ds313.3) data with a weight factor of 0.1 towards the reanalysis, for temperature and wind fields every 6 hours for the years 2000-2015.

In the standard configuration of CESM-MAM7 the vertical transport of interstitial aerosols and trace gases by deep convective clouds, use updraft and downdraft mass fluxes from the Zhang-McFarlane parameterization (Zhang and McFarlane, 1995). Currently this vertical transport is calculated separately from wet removal. Cloud-borne aerosols associated with large-scale stratiform clouds are assumed to not interact with the convective clouds.

Vertical transport by shallow convective clouds is treated similarly, using mass fluxes from the Park and Bretherton (2009) shallow convection parameterization.

We run our simulations for 16 years, from January 15th 2000 to December 15th 2015, using the CESM1.2 (CAM5) initial atmosphere state file at that date.

## 2.2-Correlative satellite data

Our simulations have been compared to satellite data from the Microwave Limb Sounder (MLS) and the Atmospheric Chemistry Experiment –Fourier Transform Spectrometer (ACE-FTS).

The MLS sounder was launched in July 2004 on-board the NASA Aura satellite. Measurements in the millimetre and submillimetre wavelength ranges are continuously made during both night and day every 165 km along the suborbital track, covering latitudes from 82° S to 82° N (Waters et al., 2006). Here, we use the MLS version 4.23 data set (Livesey et al., 2020) for CO (Pumphrey et al., 2007; Livesey et al., 2008) for selected years (2005 and 2008) and pressure levels in the UTLS. We use CO vertical profiles from 215 to 0.0046 hPa. For these pressure levels, the vertical resolution is about 5.1 km and the horizontal resolution about 570 km (at 147 hPa) (Livesey et al., 2020). The data precision is about 16 ppbv and the data accuracy is estimated at ±26 ppbv and ±30%.

The ACE-FTS instrument is an in infrared solar occultation spectrometer, providing profiles of the Earth since February 2004 from the Canadian satellite SCISAT-1 (Bernath et al., 2005). It operates in the wavelength range from 2.2 to 13.3 µm (750-4400 cm$^{-1}$) with a spectral resolution of 0.02 cm$^{-1}$. The data set provides 30 measurements per day for over 30 chemical

species from 5 km (or cloud top) up to 150 km. The horizontal weighting function of a measurement has typically a width of ~300 km. The vertical resolution is < 4 km.

## 3-Model comparison with satellite observations: CO distribution

We compare CO measurements from MLS and ACE-FTS with our simulations. While a direct comparison of aerosol extinction observations from various satellite instruments with CESM-MAM7 is not easy, e.g. due to the interference of clouds, using a trace gas (like CO) is a more straightforward approach for a comparison. In fact, three-dimensional summer distributions of CO show a distinct enhancement in the AMA and have been proved an ideal tracer to identify the AMA's location and to track the transport processes to the AMA (e.g. Park et al., 2008, Barret et al., 2016, Santee et al., 2017). The CO comparison enables a test of the model's capacity to reproduce the large-scale dynamical and morphological features, which is related to the aerosol distribution.

Figures 1a and 1b show the average summer (June-July-August) CO distribution, for the year 2008, observed by MLS in the UTLS (Fig. 1a) and produced by CESM-MAM7 (Fig. 1b), at the pressure level of 147 hPa, for MLS, and 150 hPa (average between 160-140 hPa, 3 levels), for CESM-MAM7. The locations of the general enhancement of CO mixing ratios in the AMA and of the absolute maximum above India are well reproduced by the model (i.e. they are consistent with MLS observations). It should be noted that the pressure levels used in this comparison, for CESM-MAM7 and MLS, are not exactly identical. In addition, the vertical resolutions differ as well (about 5.1 km, for MLS, and about 1 km, for CESM-MAM7). Furthermore, the temporal samplings of satellite and model data also differ: for CESM-MAM7 the temporal sampling is twice a day (noon and midnight), whereas MLS samples the Earth on distinct orbits, with a full global coverage every 3 days. Even though it is therefore possible that intensive short-time events are missed by either CESM-MAM7 or MLS, the sampling bias is not expected to present a significant source of discrepancies for 3-month averages, as shown in Figures 1a and 1b.

Compared to MLS observations, the model underestimates the CO mixing ratio by about 30%. One possible reason for this underestimation could be an underestimation of biomass burning emissions in the model (obtained from GFED4), which are a significant source for CO. We have also compared CESM-MAM7 HCN mixing ratios (a strong biomass burning tracer) with ACE-FTS HCN observations (comparison not shown here). This latter comparison shows a marked underestimation of modelled HCN amounts, which supports the hypothesis of an underestimation of biomass burning emissions. Stroppiana et al. (2010) have compared different biomass burning inventories for CO. For 2003, they found that the CO emissions range from 365 Tg (GFED3) to 1422 Tg (VGT - Vegetation Emission Inventory (CNRS-LA)) (Tansey et al., 2008), with GFED at the low end of this variability. Unlike GFED3, GFED4 include upgrades like the inclusion of small fire burned areas and a revised fuel consumption parameterization that causes global emissions to increase in comparison with the previous version. However, the effects of these adjustments vary spatially and, in particular regions like the Southeast of Asia or the North and South of Africa, the CO biomass burning emissions are lower (see Van de Werf et al. 2017). This could explain the low bias in CO mixing ratios for our comparisons with

satellite measurements. On the other hand, as mentioned in Section 2.1, the CEDS anthropogenic inventory uses a scaling process to match the CEDS emissions estimates with available inventories. In the case of anthropogenic emissions for CO, the last year from local inventories available is 2008 in Asia (from REAS) and 2010 in China (from MEIC-Multi-resolution Emission Inventory for China). As a result, the extrapolation during 2010-2015 may be an additional source of uncertainties for comparisons with observations over this period.

While reproducing monthly average features is a probing test for our simulations, catching shorter-term processes and variability is even more challenging towards the description of a complex phenomenon as the ATAL. Thus, we have also tested the model's ability to reproduce observed daily specific features. Figure 1c shows a three-day average from July 4th to 6th 2005. During this short time period, a multi-centric AMA is observed by MLS, with rather multiple maxima in eastern Asia, instead of a classical individual maximum above the Himalayan region. Our CESM-MAM7 simulations reasonably reproduce this pattern. They show a distributed pattern with maxima above eastern Asia, but also above western Asia (Fig. 1d), which is very consistent with MLS observations (Fig. 1c). For 3-day averages the sampling bias can play a significant role to explain the different patterns observed for MLS and CESM-MAM7. Therefore, some short-term features might not have been captured by the MLS instrument. Nevertheless, our simulations are very consistent with MLS observations for this short-term configuration.

We have also tested the vertical structures of CESM-MAM7 simulations, using an ACE-FTS CO mixing ratio profile in the UTLS (Fig. 1e). Observations with ACE-FTS have been chosen because of their better vertical resolution with respect to MLS. It has to be noted that the location and time of the ACE-FTS measurement profile and the model output are not exactly the same, but agree within 1° longitude, 4° latitude and within 2.5 h (see Figure 1e). The vertical distribution of CESM-MAM7 simulations shows a quite remarkable agreement with ACE-FTS observations above 200 hPa. Up to the level of ~ 400 hPa the model underestimates (as also shown for the previous examples with MLS, see Figure 1a-d) CO values by around 30%, with smaller underestimations between 400 and 200 hPa. For pressure levels lower than 180 hPa CESM-MAM7 and ACE-FTS show a remarkable consistency. Model underestimations of CO vertical concentrations have already been reported in previous studies with other models (e.g. Barret et al., 2016). The discrepancies observed between simulated and observed CO could be linked to the treatment of convection by CESM1/CAM5 together with discrepancies in emission inventories (see discussion above). In the work of He et al. (2015) underestimations of surface CO by CESM1/CAM5 have been reported especially over Asia, while the global tropospheric column of CO seems to be overestimated in their study. These authors suggest uncertainties in terms of spatial allocations of CO emissions as well as convective transport treatments. The model resolution used could also impact in the calculated transport of gases by convection. Brühl et al. (2018) have reported this fact for the transport of aerosols in their study. In our work with CESM-MAM7, we use a 1.9 x 2.5° horizontal resolution and 56 vertical levels which is an standard configuration for CESM1 and has been used in previous studies of aerosol properties (Yu et al., 2015, 2017).

Because of the sparse sampling of ACE-FTS data in the AMA, we have provided an additional comparison of the monthly vertical distribution of CO for the whole 2008 year between MLS

data and modelled CO (Fig. S1). The comparison, while showing an underestimation of the modelled vertical amounts of CO, especially below the 150 hPa level, presents spatial distributions of CO which are in good agreement.

The comparison of simulated CO with observed MLS and ACE-FTS CO in the UTLS allows us to conclude that, except for a possible underestimation of CO emissions, the model is able to reproduce the position and spatial extent of the Asian monsoon anticyclone in our simulations.

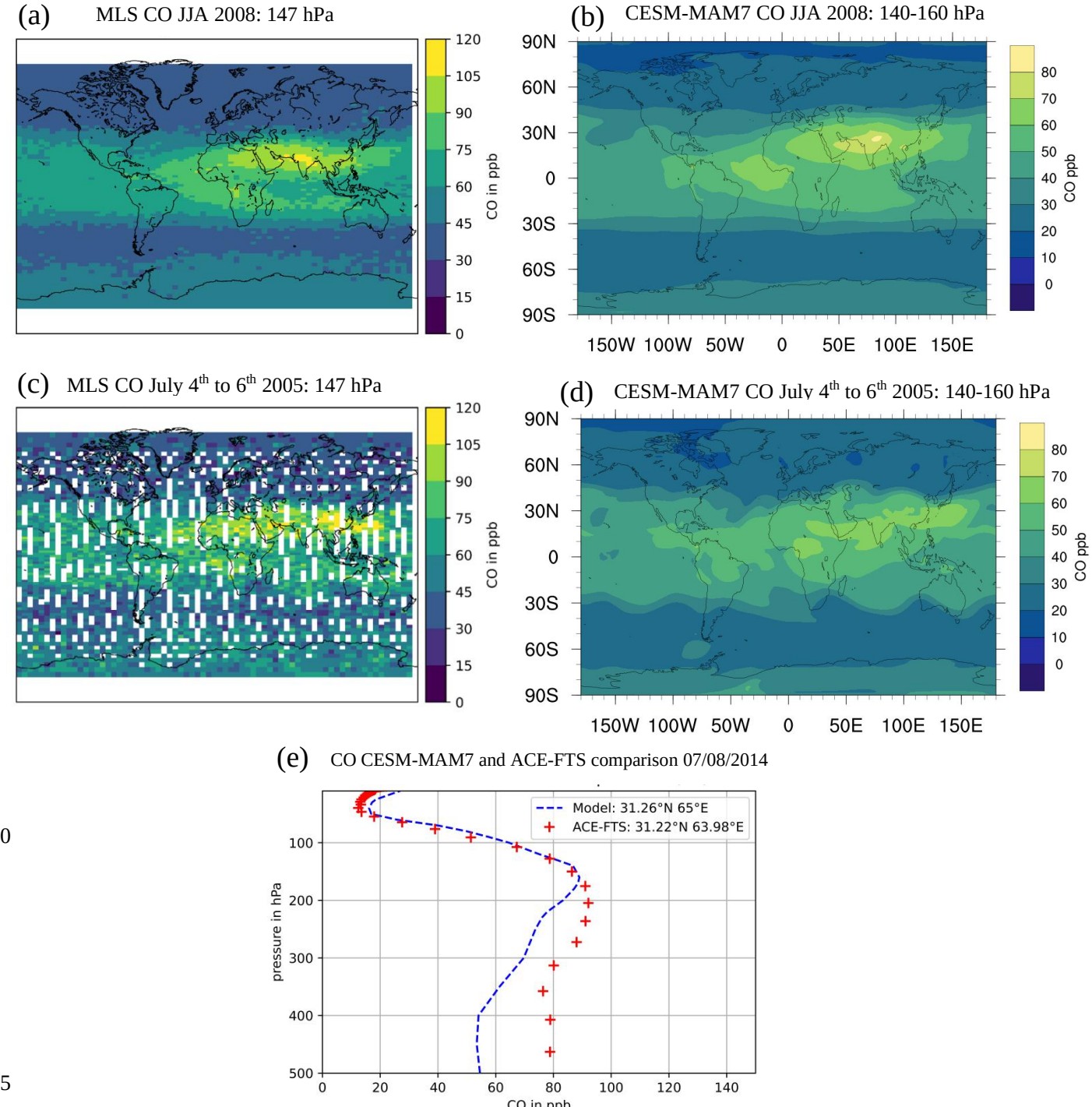

**Figure 1: (a) Average MLS CO mixing ratio distribution for June-July-August 2008 at 147 hPa**

**pressure level and (b) average CESM-MAM7 CO mixing ratio distribution for June-July-August 2008 between 140 and 160 hPa. (c) 3-day average for the MLS CO mixing ratios at 147 hPa (July 4th to 6th 2005) and (d) respective CESM-MAM7 simulations, for July 4th to 6th 2005 between 140 and 160 hPa. (e) ACE-FTS and CESM-MAM7 vertical CO profiles for Aug 7th 2014 at 31.22°N - 63.98°E, 14:30 UTC and 31.26°N, 65.00°E, 12:00 UTC, respectively.**

## 4 - Results and Discussions

### 4.1 - Aerosol distribution and composition

Figure 2 shows the CESM-MAM7 regional distribution, over an extended area centered around the AMA region, of different aerosol types: sulfate, SOA, POM, BC, ammonium and mineral dust. The accumulation mode (a1) is here shown for all aerosol types, except for mineral dust (for which Fine Soil Dust mode (a5) is shown). These maps represent average aerosol concentrations, for July-August 2014, at three different pressure levels: 120, 100 and 80 hPa, respectively (approximately 15.0, 16.5 and 18.0 km). Concentrations of sea salt particles, also modelled in our study, are negligible and therefore are not shown in Fig. 2. The model reproduces the horizontal distribution of the ATAL, i.e. an increase in aerosol concentration in the AMA region with elevated aerosol concentration at 120-100 hPa (upper troposphere) and noticeably decreasing for pressures lower than 80 hPa (altitudes higher than 18.0 km, lower stratosphere).

Figure 2 shows that dust is the principal aerosol species in the ATAL, in terms of mass concentration, in our simulations. These results agree with some previous modelling studies (e.g. Fadnavis et al., 2013;Ma et al., 2019). Our results show an aerosol dust concentration at 100 hPa of about 100 ng/m$^3$ in agreement with the findings of Ma et al. (2019) who have reported a value > 100 ng/m$^3$ at 16 km with ECHAM/MESSy. Fadnavis et al. (2013) using the ECHAM5-HAMMOZ model have simulated a value of ~ 30 ng/m$^3$ for dust, similarly to Fairlie et al. (2020) who have reported a concentration of ~ 20 ng/m$^3$ using the GEOS-Chem model. According to Lau et al. (2018), high burdens of dust are found in the ASM region, transported from the desert regions which are trapped by local topography and accumulated to high concentration over the southern and eastern foothills of the Tibetan Plateau and transported to the ATAL (~12-16 km) region by increased vertical motion associated with deep convective motions. It is not clear if processes that drive convection and have an impact on its modelling (convective schemes, model resolutions, reanalyses used to nudge the models), accounted for in the above-mentioned model studies, can explain the differences in terms of simulated amount of dust. For instance, Brühl et al. (2018) have shown that the amounts of dust reaching the UTLS region in the EMAC model are sensitive to model resolution, showing that a resolution of 1.88 x 1.88° and 90 vertical levels has the best fits with spaceborne observations of dust extinction. In our work with CESM-MAM7, we use a 1.9 x 2.5° horizontal resolution and 56 vertical levels which is one of the standard configurations for CESM1 and has been used in previous studies of aerosol properties (e.g. Yu et al., 2015, 2017). These resolutions are lower

than those in Brühl et al. (2018) and this could impact the amount of dust reaching the UTLS in CESM-MAM7 as result of differences in convection top height and overshooting convection.

In addition, Wu et al . (2019), using CESM1-CAM5 with the default scheme for the dust emissions (Zender et al., 2003), same scheme used in the present study, have shown that the model overestimates dust extinction over the Taklamakan and Gobi deserts during the summer period. Such high biases in dust extinction have been attributed to excessive convective transport, lack of secondary activation of aerosol entrained into convective updrafts and strong dust transport in the upper troposphere from Africa and the Middle East. These hypotheses,

together with differences in the model resolution, could explain the higher dust amounts in our CESM-MAM7 simulations, which use the same default scheme for the generation of dust.

The discrepancies observed between different models could also result from the different schemes used for the dust lifting, as well as the sensitivity of dust release to surface conditions, particularly to surface winds and soil properties.

Other main aerosol components contributing to the ATAL in our simulations, are sulfates, followed by SOA, POM, ammonium and to a lesser extent BC. Yu et al. (2015), using CESM1/CARMA model, have suggested that the ATAL (at altitudes levels between 230-100 hPa) is principally composed of organics (~60 %) and sulfates (~ 40%), while an aerosol enhancement due to dust above Africa was also observed. Fadnavis et al. (2013) have found

that dominating aerosol types in the ATAL are dust and sulfates, followed by organic carbon and BC aerosols. Fairlie et al. (2020) have also simulated that sulfate and primary organic aerosols are major components of the ATAL but, as in the work of Gu et al. (2016), with nitrate as the predominant aerosol.

As discussed in previous studies, the spatial extent, strength and position of the AMA is highly

variable due to the dynamical seasonal variability of the ASM (e.g. Randel and Park, 2006; Garny and Randel, 2013; Lau et al., 2018; Basha et al., 2019). Due to this dynamical variability the tracer concentrations are strongly controlled by the oscillations and shedding of the AMA, that therefore affect the ATAL extent and composition. In order to determine the aerosols burden within the ATAL we have defined a simple criterion to isolate the ATAL horizontal extent,

i.e. where there is a high probability to find AMA air masses, based on a threshold on the geopotential height (GPH) values. Similar empirical selections of high GPH values to represent anticyclone boundaries have been used in a number of previous works, e.g. Highwood and Hoskins (1998), Bergman et al. (2013), Barret et al. (2016), Pan et al. (2016). For the subsequent analysis, we identify the AMA region based on GPH values higher than 16.7 km at

100 hPa. Based on this criterion, a wide region from around 20-130 °E and 20-45 °N is generally selected. Then, we define a static box corresponding to the highest probability to find air masses delimited by the anticyclone. According to these considerations, we have finally chosen to restrict the box to 20-35 °N and 60-100 °E to identify and study the ATAL composition (blue box in the central panel of first row in Fig. 2).


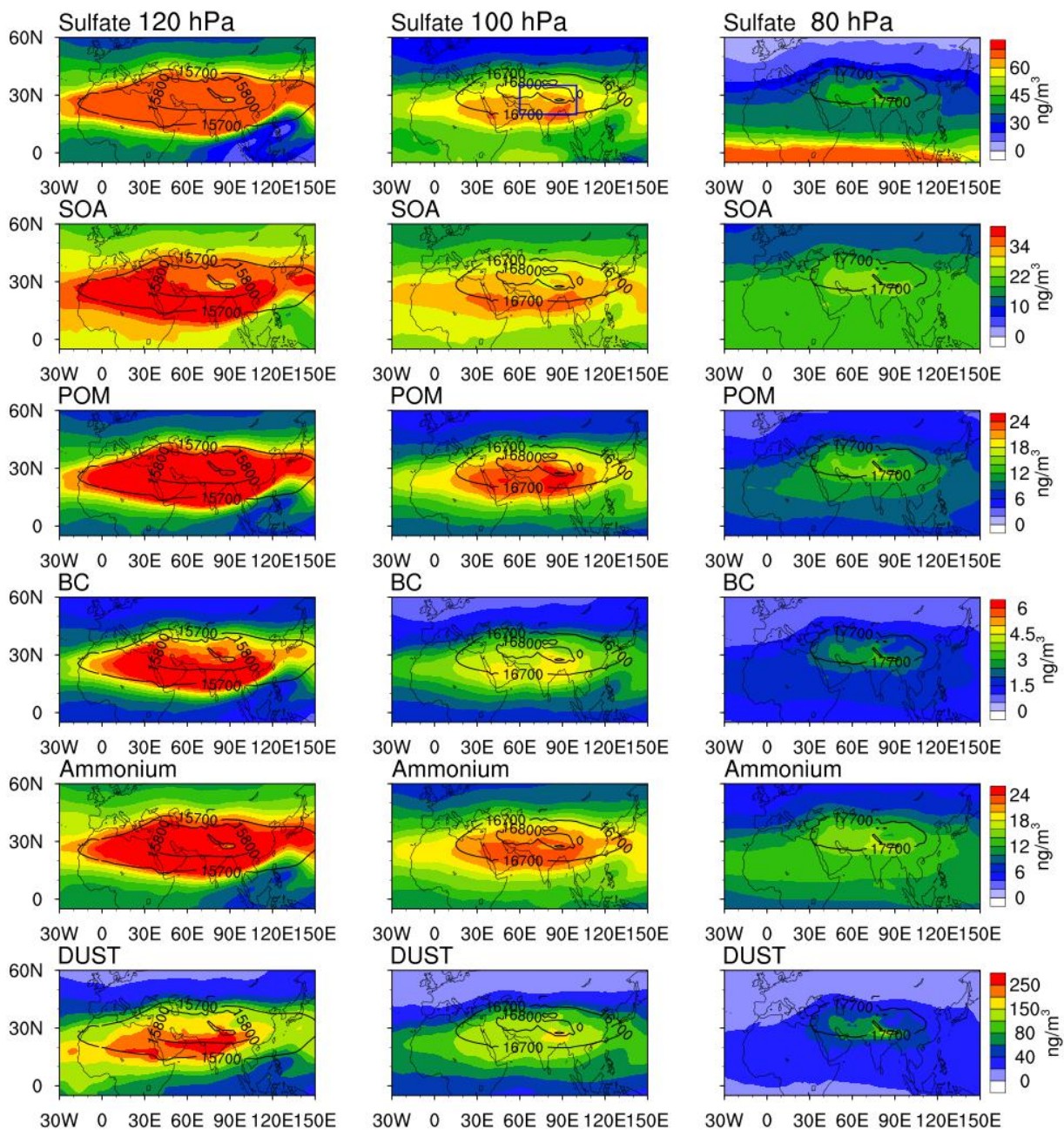

**Figure 2: Spatial distribution of the aerosols mass concentration, averaged over July-August 2014, from CESM-MAM7 simulations, for six different aerosol types. From top to bottom row: sulfate, SOA, POM, BC, ammonium (in the accumulation mode) and mineral dust (in the fine dust soil mode). From left to right column: 120, 100 and 80 hPa pressure levels. Note the different color scale ranges. The black lines in the map represent the geopotential height > 15700 m at 120 hPa, >16700 m at 100 hPa and > 17700 m at 80 hPa. The blue box (2nd panel) represents the area chosen for the subsequent ATAL-specialised analyses (20-35 °N, 60-100 °E).**

## 4.2 - Vertical distribution of the ATAL

In Fig. 3 we show CESM-MAM7 vertical aerosol mass mixing ratio profiles for the accumulation mode, averaged from June to August within the blue box of Fig. 2, for two selected years, 2000

and 2014. Our focus on the accumulation mode is justified by the fact that it is the principal mode that contributes to the ATAL (see Fig. S2 in Supplement), with mostly anthropogenic origin. In this first analysis, we have excluded dust. Dust is still the most important ATAL component, in our simulations, in terms of mass, but its burden and variability are mostly subject to natural factors and their variability.

A vertical region with marked localized increase of the concentration of all the aerosols types is observed between 300 and 80 hPa. This is what is expected as a manifestation of the ATAL, as it is broadly the vertical region where the AMA is located. The vertical structure of the AMA-related dynamics has been investigated by several authors (e.g. Park et al., 2009; Bergman et al., 2013; Garny and Randel, 2013; Brunamonti et al., 2018; Bian et al, 2020), showing evidence of deep convection and confinement extending up to 1.5–2.0 km above the cold-point tropopause. Enhanced aerosol backscatter also reveals the signature of the ATAL over the same altitude range (Vernier et al., 2015; Brunamonti et al., 2018). This location suggests that the existence of the layer is tied to a large-scale vertical transport in the anticyclone, i.e around 200 to 80 hPa (~13 to 18 km) depending on the location and time.

Our simulations show a characteristic "double-peak" vertical configuration with two relative maxima, one at higher altitudes (~80-120 hPa) and the other at lower altitudes (~200-300 hPa). During early phases of the ASM (e.g., June, Fig. 3a) the maximum of aerosol concentrations is generally located between 200 and 80 hPa; later on (e.g., July and August, Figs. 3c,e) an aerosol enhancement at lower altitudes (around 250 hPa), superimposed with a maximum at around 100 hPa, is found. This "double-peak" vertical structure could be explained looking at the interplay of interstitial and in-cloud aerosols in CESM-MAM7. As was detailed in the Sect. 2.1, the interstitial aerosols include both clear-sky/dry aerosols and aerosols contained within convective clouds. Our simulations show that during the mature phase of the AMA (July and August), at the same time of increased convection, the AP in convective clouds (maximum of convective outflow at ~ 250 hPa) also increase. This causes a maximum of aerosols at lower altitudes. Figure S3 shows the cloud ice fraction for 2014 averaged for the blue box. In June the fraction of clouds is much smaller than in July and August.

This "double-peak" vertical structure can be found in some observations from recent aircraft and balloon campaigns but not discussed. For instance, particle counting observations during the 2015 BATAL campaign (Vernier et al., 2018) have shown two maxima in the aerosol concentration profile, at ~ 17 km and ~ 14-15 km (See Fig. 11 in that paper). They mainly associate the enhanced aerosol concentrations with the influence of convective transport of regional Indian pollutants and the observed lower peak with the presence of ice particles . During the StratoClim campaign carried out in August 2016 and 2017, Brunamonti et al. (2018) have observed the frequent presence of ice particles in the AMA, often found embedded within the ATAL. They have shown a clear-sky/dry aerosol ATAL signal between 70 and 150 hPa after the application of a cloud filter. As another example of this "double-peak" feature in the vertical ATAL profile, Höpfner et al. (2019) have observed two peaks for ammonium nitrate aerosols on July 2017 during the StratoClim campaign (see Fig 4 in this paper). These results support our hypothesis about the simulated lower peak associated with particles in convective

clouds or in the convective outflow, although the occurrence of such lower-peak feature needs confirmation from further in situ observations.

We have also tried to separate the overall in-cloud and the purely dry aerosols (these latter likely coming from nucleation/condensation processes). In order to analyze the contribution of dry aerosols to the ATAL we have carried out an analysis to reduce the contribution from convective cloud-borne aerosols. For this purpose, we have filtered out the profiles, in our blue box, for which the extinction coefficient is larger than an arbitrary threshold ($1.0\ 10^{-3}$ km$^{-1}$ in our case). Figure S4 shows the evaluation of different filters for the extinction coefficient applied for our box domain for August 2014 (same behavior is observed for July, not shown). We have applied a filter of $8.0\ 10^{-4}$, $9.0\ 10^{-4}$, $1.0\ 10^{-3}$ and $2.0\times10^{-3}$ km$^{-1}$, respectively and have evaluated the maximum value obtained at around 100 hPa where our upper peak is located. By varying these threshold values, we arrive to the point of isolating the upper peak, which is satisfactory for $1.0\ 10^{-3}$ km$^{-1}$. Figures 3b,d,f show the vertical aerosol profiles with the applied filter, from where an isolated upper peak can be seen. This peak, due to the mentioned filtering, is associated with aerosols with limited radiation extinction. Large extinction values are associated with in-cloud aerosols, which are larger in size due to liquid phase formation, freezing and/or hygroscopic growth (depending on the primary or secondary nature of aerosols). We then identify as clear sky/dry AP the ones associated with this upper peak (120-80 hPa). The comparison with AP vertical profiles from Fig. 3a,c,e allows us to point out that in CESM-MAM7 both types of aerosols contribute to the ATAL, i.e. clear-sky/dry aerosols and convective cloud-borne aerosols.

It is worth noticing that, for these two selected years (2000 and 2014), the aerosol profiles can differ from one aerosol type to another but are quite similar for a given month/year, and a double- or single-peak structure is observed. This variability observed in the ATAL's vertical profiles also reflects the aspect of the dynamical variability of the AMA, which can be put in relationship with both the long-range transport and convection, as it was shown in previous studies (e.g. Qie et al., 2014; Pan et al., 2016; Santee et al., 2017).

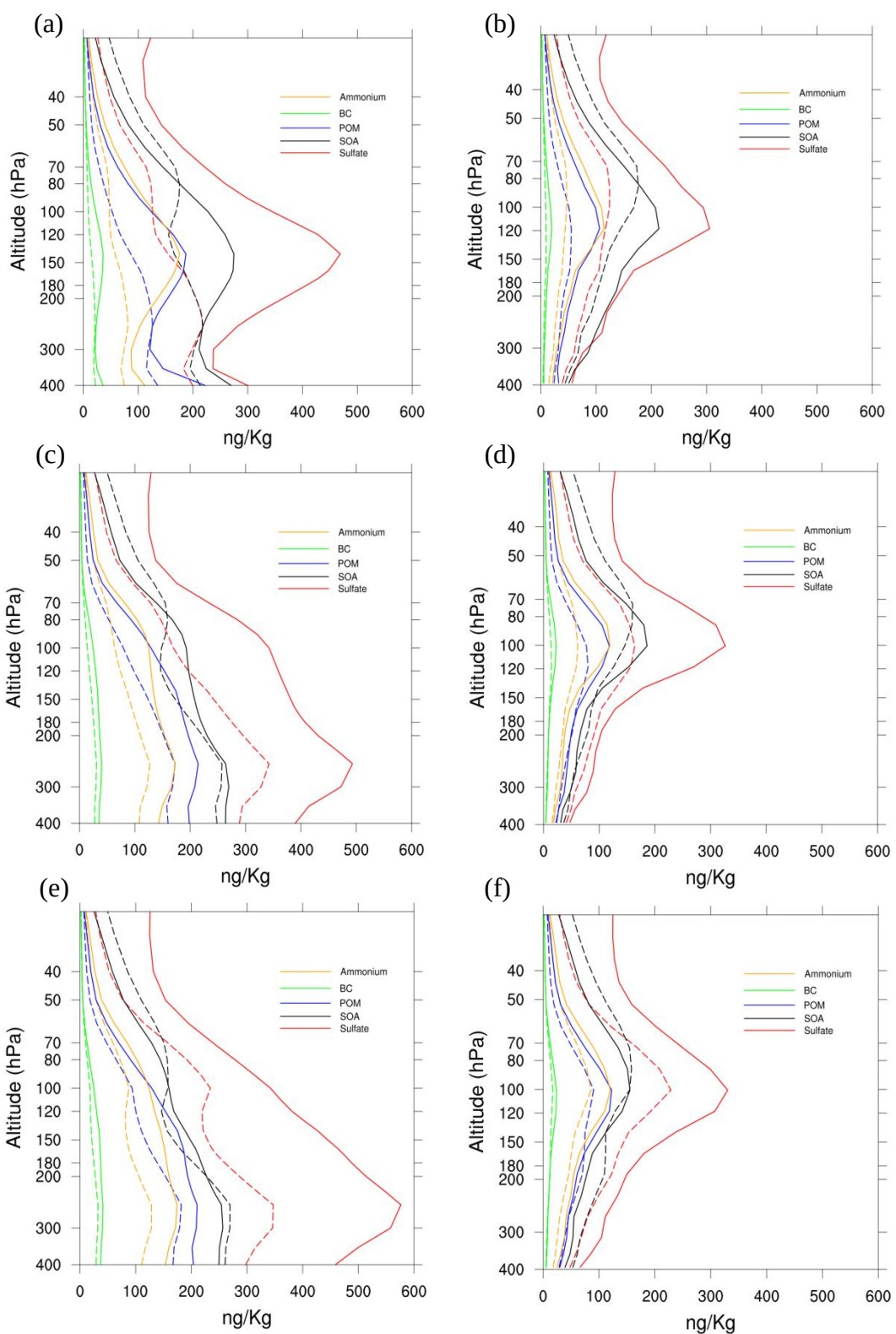

530

**Figure 3: Modelling vertical profiles of aerosol mass concentration of sulfate, SOA, POM, ammonium and BC in the accumulation mode (a1) in ng/kg averaged between 20-35 °N and 60-100 °E, the dash line correspond to the year 2000, solid lines the year 2014. (a) profile for June (c) July and (e) August. (b), (d), (f) same as (a), (c) and (e) but with the extinction filter**

535 **applied (>1.0 x10⁻³ km⁻¹) to reduce the contribution of convective cloud-borne aerosols.**

### 4.3 – Trends in aerosol composition of the ATAL

Figures 4a-d shows the annual average aerosol total mass concentrations for all the aerosol types simulated by CESM-MAM7, in the period 2000-2015, for all modes (Figs. 4a,c) and the isolated accumulation mode (Fig. 4b,d). To account for the whole double-peak phenomenology and to isolate the single dry AP peak (see discussion in Sec. 4.2), the concentrations are averaged between 200-80 hPa (Fig. 4a,b) and 120-80 hPa (Fig. 4c,d). These two vertical ranges allow the differentiation of the ATAL composition based on in-cloud processes or, from another point of view, to describe how the composition changes depending on the altitude. No filter has been applied to show the contribution of all aerosols.

The aerosol type that dominates the ATAL, for both altitude ranges, is dust, followed by sulfates and organic particles (secondary and primary). The comparison between Fig 4a and 4c shows that at higher altitudes the amount of sulfates increases slightly and, more markedly, dust amount decreases. Figure 4e shows the percent contribution of aerosols types to the ATAL, between 120-80 hPa. It is evident that although less dust reaches higher altitudes, this aerosol type is still the mass-dominant aerosol type in the ATAL, contributing around 60%. Even if there still is a large disagreement among reported studies about the exact amount of dust present in the ATAL, it is clear that in our study this natural component contributes significantly to the ATAL seasonal build-up due to its transport from the nearby desert regions, like Taklamakan and Thar deserts, and the northern slope areas of Tibetan Plateau (Lau et al., 2018, Ma et al., 2019). As was mentioned in the Sec 4.1 the difference in the amount of dust reported by the different authors may be related to the different schemes used for the generation of dust, e.g. how the topography is represented in the model, the resolution of the model and the parameterization of the convection processes.

Wu et al. (2018) have evaluated dust emissions in East Asia simulated by 15 climate models participating in the Coupled Model Intercomparison Project Phase 5 (CMIP5) during 1961-2005. They have found discrepancies with the observations for all the models, because climate models may not sufficiently represent the trends of surface wind speeds and precipitation. This indicates that there is still a need to improve the representation of the dust cycle in climate models to simulate long-term dust changes.

With the intention to analyze the composition of the ATAL in terms of anthropogenic and biomass burning emissions we discuss more in detail the contribution of the non-dust aerosols, for which the accumulation mode at two different altitude ranges is shown in Fig. 4b, d. Excluding dust particles, the accumulation mode (a1, size range: 0.056-0.26 µm) is the principal mode that contributes to the ATAL. This can be seen in the Fig. S2 in the Supplement. Hence, anthropogenic and biomass burning aerosols that reach the ATAL are principally small and young. The same behavior is observed in the 200-80 hPa range (Figure not shown here).

Sulfate aerosols from moderate-to-strong volcanic eruptions, with injection in the UTLS, can also interact with the dynamical features of the AMA (e.g. Sellitto at al., 2017) and, under certain conditions, can impact the ATAL aerosol population. Larger sulfate concentrations in 2009 and 2011 are linked to the volcanic eruptions of Sarychev (June 2009) and Nabro (June 2011). These eruptions injected large quantities of $SO_2$ into the UTLS, just before the onset of

the AMA. The subsequently formed volcanic sulfates from $SO_2$ conversion to particles rise the background inside and outside the AMA and therefore contributed to the ATAL burden, during these two years. For these years influenced by moderate volcanic eruptions the concentration of sulfate increases drastically and reaches or even exceeds the dust concentration (see Fig. 4). Excluding dust and focusing on the mostly anthropogenic accumulation mode, Fig. 4f suggest that the fraction of the ATAL of anthropogenic origin is composed of about 40% sulfate, 30% SOA, 15% POM, 14% ammonium and less than 3% BC. Compared to the results reported by Yu et al. (2015), our results show about the same percentage of sulfate in the ATAL but less organics, i.e ~45% aggregating SOA and POM for our study compared with 60 % of organic as reported by Yu et al. (2015).

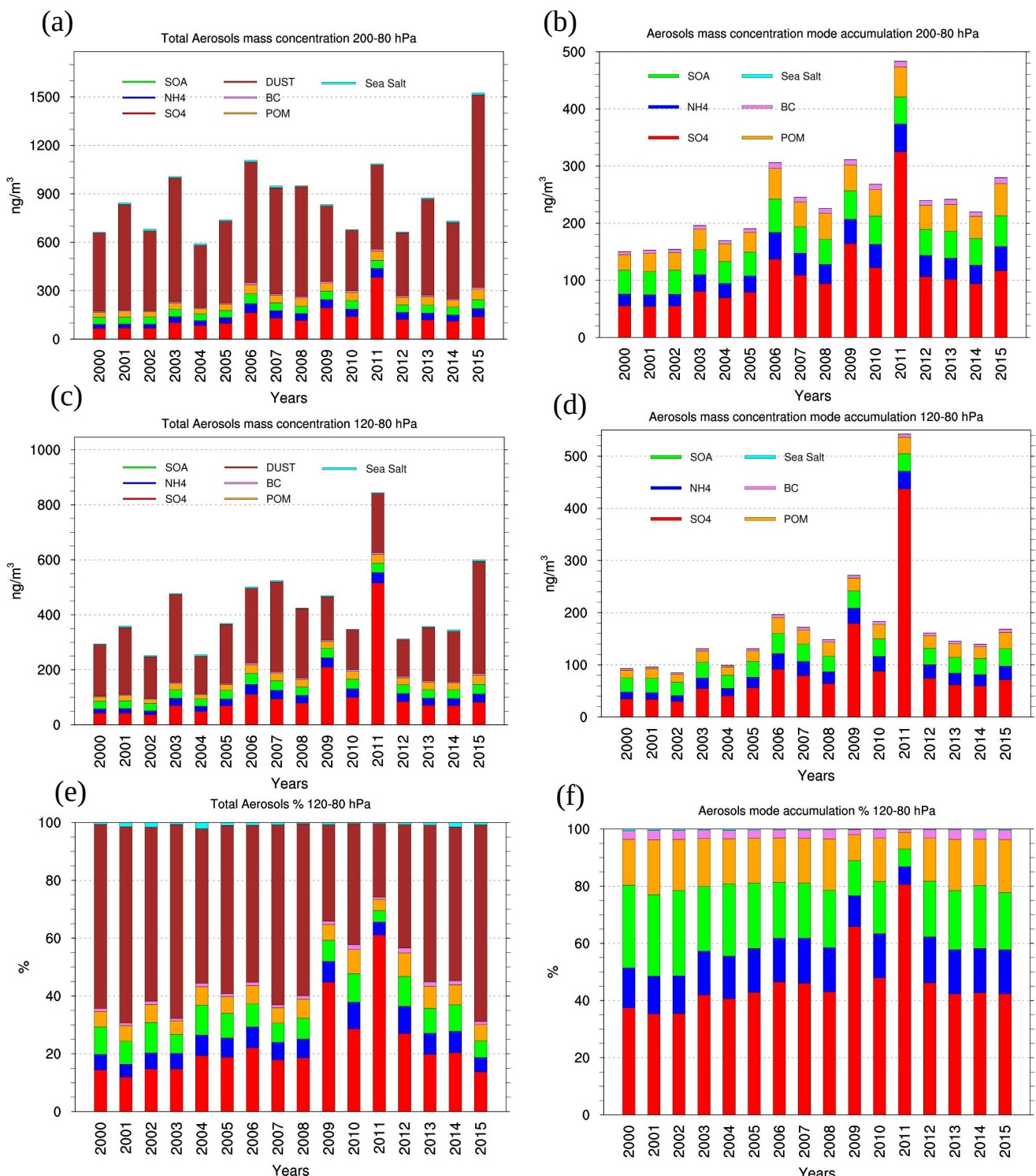

**Figure 4: Evolution of the total aerosol mass concentration of all the aerosol types present in CESM-MAM7 in all the modes averaged at 20-35°N, 60-100 °E for July-August, (a) between 200-80 hPa, (c) between 120-80 hPa , (e) percent amount at 120-80 hPa. Panels (b), (d) and (f) are the same as panels (a), (c) and (e) but only for the aerosols in the accumulation mode.**

In the following, we evaluate the decadal trends of the different aerosol types in the ATAL. In particular, we have estimated the trends for the dust in the fine soil dust mode (Fig. 5a) and all other aerosol types in the accumulation mode (Figs. 5b,c). The concentrations for each year are averaged between 120-80 hPa pressure levels and over the domain defined by the blue

box of Fig. 2, excluding the years with volcanic eruptions impacting the UTLS, i.e. 2005 to 2009 and 2011-2012 (Manam: April 2005; Soufrière Hills: August 2006; Tavurvur: October 2006; Okmok: August 2008; Kasatochi: August 2008; Sarychev: June 2009; Nabro: June 2011, taken from Khaykin et al. 2017, see Table 3 in their paper).

As can been see from Fig. 5a, dust does not display any clear trend. The p-value (a p-value less than 0.05 confirms that a statistical test is significant in indicating strong evidence against the null hypothesis) of 0.64 confirms an insignificant positive value (same behaviour is observed in the 200-80 hPa range, figure not shown here), reinforcing the evidence that the variation in dust concentration in the ATAL region is only subject to the natural interannual variability, as pointed out in Yuan et al. (2018), with no specific long-term trends. The sparse variations of dust in the ATAL reflects the influence of other factors not related to the ASM, like the variability of extratropical westerlies that can strongly affect the long-range dust transport at high elevations, or the wet scavenging in and below clouds that can overcome the effect of lofting by deep convection.

Figures 5b,c show the trends for all the aerosols in the accumulation mode averaged in our box over the 120-80 hPa vertical level range, respectively without and with the extinction filter applied so as to isolate dry from in-cloud (including from convective clouds) aerosols. All the aerosol types show an increase over the simulated 16-years. This mirrors the increase of the emissions in Asia. From Fig. 5b, it can be seen that sulfate aerosols trends in the ATAL, roughly doubling their concentration from ~36 $ng/m^3$ in 2000 to ~75 $ng/m^3$ in 2015 (i.e. about 108% increase in 15 years). Marked increases are also observed for POM (~80%), ammonium (~100%) and BC (~93%), while for SOA the trend is weaker, i.e. going from ~ 27 to 33 $ng/m^3$ (~24%). The concentrations for the years 2000 and 2015, the percentage of increment for the 15 modelled years, the R coefficient for the trends and p-value are summarized in Tab. 2.

Figure 5c shows the trends of dry aerosols, i.e. with the extinction filter applied, in the ATAL between 120 and 80 hPa. The comparison between Fig. 5c with 5b, together with the values reported in Tab. 2, show that the increasing trends and correlation values are slightly smaller than values reported without applying the filter. This reflects the fact that at 120-80 hPa the dry aerosols contribute to a larger fraction of the ATAL than convective cloud-borne aerosols.

The analysis of differences without and with the application of the extinction filter (i.e, (dry + convective) - (dry) aerosols) reveals that the increase for convective cloud-borne aerosols between 120 and 80 hPa in our box domain is ~22% for sulfate, ~10% for SOA, ~28% for POM, ~ 20% for $NH_4$ and ~ 25% for BC (values derived from Tab. 2).

We have also carried out the same analysis for the larger altitude interval of the ATAL, i. e. between 200 and 80 hPa (Fig. 5d and e). More convective cloud-borne aerosols are present in this case. Thus, the differences for the cases without versus with the extinction filter (calculated from Tab. 2) are larger than the previous case (~ 36% for sulfates, ~44% for POM, ~32% for $NH_4$, 47% for BC and ~ 21% for SOA).

| Aerosol | SO$_4$ | SOA | POM | NH$_4$ | BC | DUST | SO$_4$ | SOA | POM | NH$_4$ | BC |
|---|---|---|---|---|---|---|---|---|---|---|---|
| **120-80 hPa** | Without Filter | | | | | | Filter Extinction 1 x10$^{-3}$ km$^{-1}$ | | | | |
| **2000 (ng/m³)** | 36 | 26.8 | 16.4 | 13.4 | 2.9 | 159 | 35.7 | 26.3 | 16 | 13.2 | 2.8 |
| **2015 (ng/m³)** | 75 | 33.4 | 29.4 | 26.7 | 5.6 | 188 | 66.3 | 30 | 24.2 | 23.7 | 4.7 |
| **% increment** | 108.3 | 24.6 | 79.3 | 99.2 | 93.1 | 18.2 | 85.7 | 14 | 51.2 | 79.5 | 68 |
| **R coefficient** | 0.78 | 0.79 | 0.85 | 0.83 | 0.86 | 0.18 | 0.72 | 0.63 | 0.80 | 0.79 | 0.80 |
| **p-value** | 0.010 | 0.010 | 0.003 | 0.005 | 0.002 | 0.64 | 0.02 | 0.07 | 0.007 | 0.01 | 0.008 |
| **200-80 hPa** | Without Filter | | | | | | Filter Extinction 1 x10$^{-3}$ km$^{-1}$ | | | | |
| **2000 (ng/m³)** | 53 | 37.7 | 26.5 | 19.6 | 4.7 | | 39 | 27.6 | 18.3 | 14.4 | 3.2 |
| **2015 (ng/m³)** | 108 | 47.5 | 46.3 | 37.9 | 9 | | 65.2 | 28.8 | 23.9 | 23.2 | 4.6 |
| **% increment** | 103.8 | 26 | 75 | 93.4 | 91.5 | | 67 | 4.3 | 30.6 | 61 | 44 |
| **R coefficient** | 0.87 | 0.84 | 0.88 | 0.88 | 0.89 | | 0.70 | 0.27 | 0.74 | 0.74 | 0.74 |
| **p-value** | 0.002 | 0.003 | 0.001 | 0.001 | 0.0008 | | 0.04 | 0.48 | 0.02 | 0.02 | 0.02 |


**Table 2: Averaged aerosol mass concentration and percentage of the increase from 2000 to 2015 for SO$_4$, SOA, POM, NH$_4$ and BC, averaged for the summer period July-August at 20-35 °N, 60-100 °E between 120-80hPa and 200-80 hPa, without and with the extinction filter applied. R coefficient from Fig. 5b to 5e and the respective p-value are also reported.**


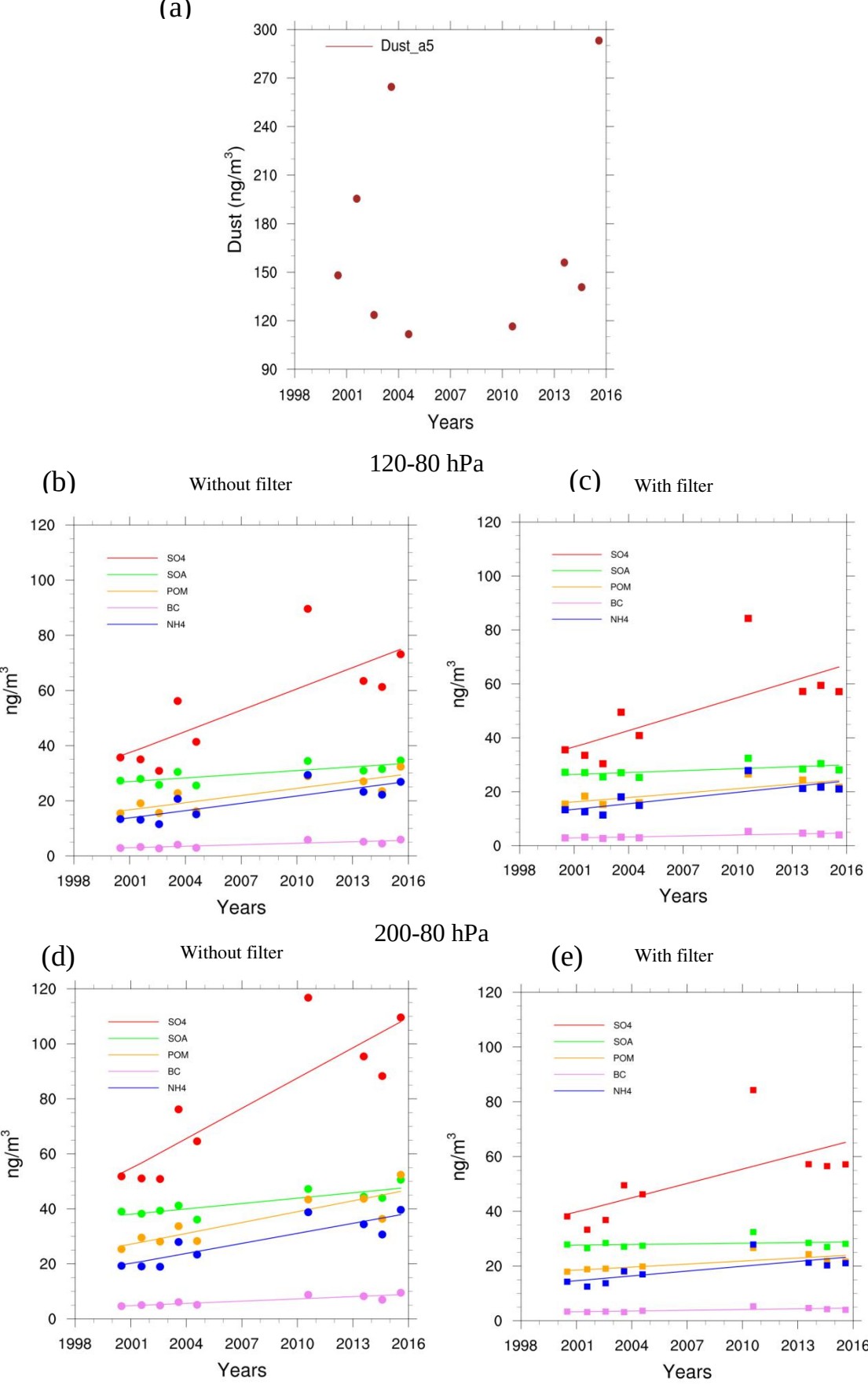

**Figure 5: Aerosol mass concentration trends simulated by CESM-MAM7 averaged between 20-35 °N, 60-100°E for July and August. (a) for dust in the Fine Soil Dust mode between 120-80 hPa  (b) respectively for SO₄, SOA, POM, BC, NH₄ in the accumulation mode between 120-80**

**hPa (c) Same as (b) but with the extinction filter applied. (d) and (e) same that (b) and (c)**
**but averaged between 200-80 hPa. The plots show the trends excluding the years with**
**volcanic eruptions impacting the UTLS.**

## 4.4 - Aerosol Optical Depth (AOD) of the ATAL

Fig. 6a and 6b shows the aerosol optical depth (AOD) at 550 nm averaged for July-August and
between 20-35 °N latitude, for selected years between 2000 and 2015, as function of the
longitude. As done before, two different altitude ranges, 200-80 hPa (Fig. 6a) and 120-80 hPa
(Fig. 6b), are analyzed, to account for the double-peak ATAL introduced in Sect. 4.2. The AOD is
calculated from the total aerosol extinction provided by CESM-MAM7. Then, in the AOD the
extinction of all the aerosols from all modes and both dry aerosols and convective in-cloud
aerosols are taken into account in the AOD. For the full double-peak ATAL (20-80 hPa), AOD
values from about 0.007, in 2000, to about 0.016, in 2015, are obtained in the core of the AMA
region (Fig. 6a). These values are about a factor 2-3 larger than the values reported by Vernier
et al. (2015) using SAGEII and CALIOP satellite data. The values reported by Vernier et al. (2015)
include a cloud-screening procedure to attempt to remove cirrus clouds. One can argue that
this filter might have screened out some aerosols with high extinction, like those we identify
from convective cloud-borne aerosols in our lower peak. Vernier et al. (2015) have also used a
depolarization filter which might have removed irregularly-shaped particles, with a possible
impact on dust. This possibility has been suggested by Yu et al. (2015) who have also reported
an AOD simulated by the CESM1/CARMA model with a factor of ~2 larger than Vernier et al.
(2015). The maximum observed in Fig. 6a are comparable with those of Yu et al. (2015) despite
the fact that we have used a different latitudinal extent (15° to 45° N) to study the ATAL.
AOD values from about 0.0019, in 2000, to about 0.004, in 2015, are found over the 120-80
hPa range where dry aerosols dominate (Fig. 6b). Between 200 and 80 hPa, higher AOD values
are obtained as result of a large contribution of convective cloud-borne aerosols at this altitude
range. The difference between the AOD values obtained for the two altitude ranges in Fig 6a
and 6b points at the importance of what we have identified as convective in-cloud aerosols.
Fig. 6c and 6d shows the temporal evolution of the yearly ATAL AOD from 2000 to 2015 for our
selected box (20-35 °N, 60-100°E), for the 200-80 hPa (Fig. 6a) and 120-80 hPa (Fig. 6d)
vertical ranges. Our simulated trend is comparable to that observed by Vernier et al (2015)
with an increase of a factor ~1.5-2.0 over the period although AOD trend values are very
difficult to compare between both works due to different considered periods and different cloud
filtering procedures. Fig 6c,d show that accounting for the double-ATAL-peak structure leads to
different AOD trend values and reflects the importance of the altitude range used to estimate
the year-to-year variability.
The attribution of the possible causes to the increase of the aerosol content and optical depth
in the ATAL between 2000 and 2015 (e.g. increase in Asian emissions, more efficient vertical
transport or different chemical/microphysical processes) requires further investigations and the
continuous monitoring of ATAL burden and properties in the future.

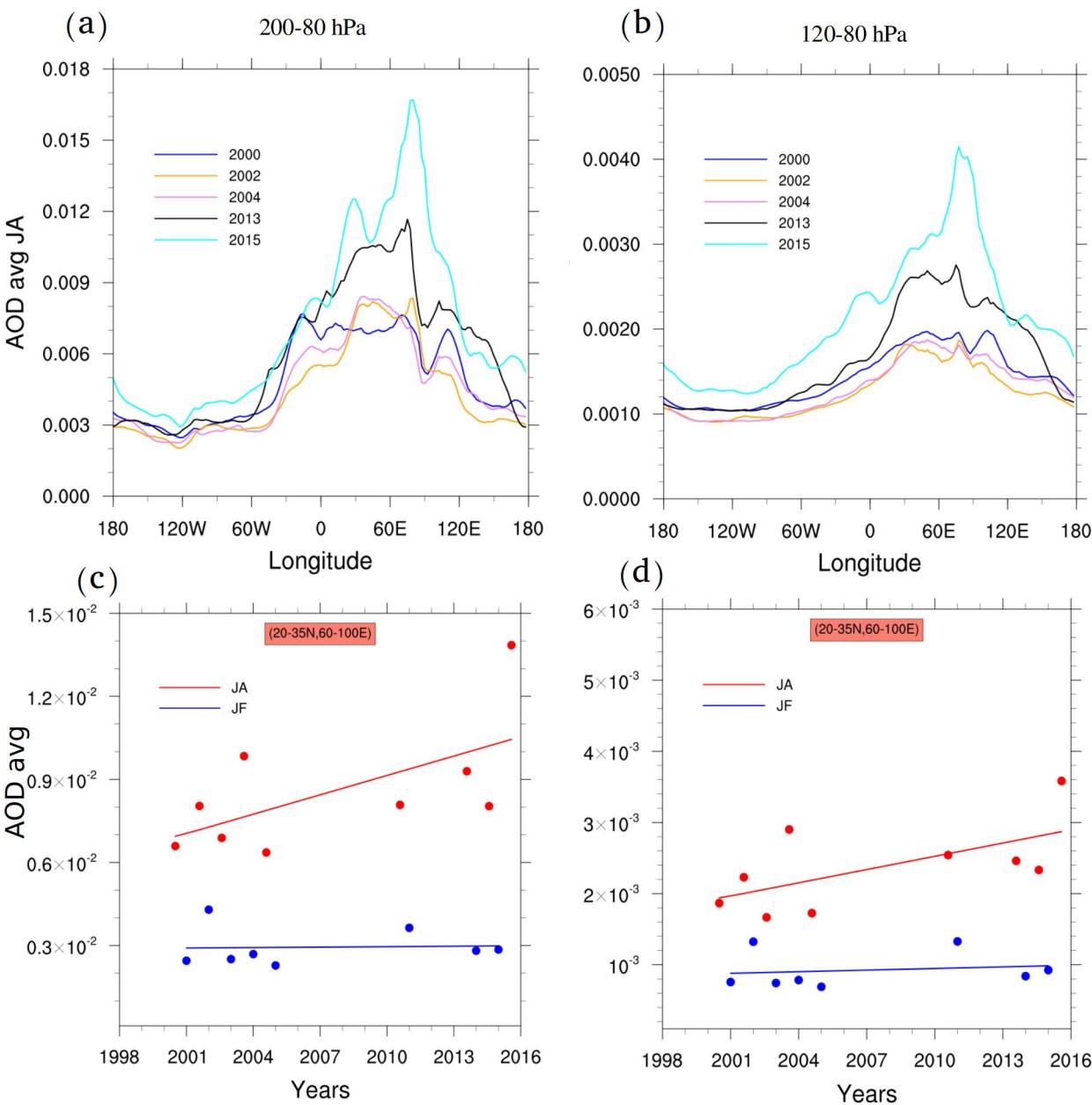

**Figure 6: Upper panels: AOD at 550 nm averaged from 20-35 °N of latitude for July-August (a) between 200-80 hPa (around 13 to 18 km) (b) between 120-80 hPa (around 15.7 to 18 km). Different colors represent the different selected years. Lower panels: AOD trends for July-August (red) and January-February (blue) averaged between 20-35 °N and 60-100 °E (c) between 200-80 hPa and (d) between 120-80 hPa. The plots show the trends excluding the years with volcanic eruptions impacting the UTLS.**

## 5-Conclusions

In this paper, we have presented the results for our long-term simulation, i.e. 16 years (January 15th 2000- December 15th 2015), to investigate the composition and trends of the specific

ATAL aerosols using the CESM-MAM7 model. The model was driven by the CMIP6 emissions inventory for the anthropogenic and biomass burning emissions of the principal trace gases and aerosols, while the biogenic emissions were taken from the MEGAN-MACC inventory. During summer, a confinement of polluted air masses has been found within the AMA region, which is tied to the ATAL position. The model results show overall good agreement with the space-time behaviour of CO in the UTLS region observed by the MLS and ACE-FTS space-borne instruments, despite a possible underestimation in the CO burden due to the underestimation of surface emissions. In particular, the horizontal distribution of modelled CO is in good agreement with MLS data and the vertical structure in the AMA shows a maximum near 150 hPa in agreement with the available ACE-FTS observations.

Our model results indicate that dust is a dominating aerosol type in terms of mass in the ATAL in agreement with other studies (e.g. Lau et al. 2018, Ma et al., 2019). However, the lack of in situ or satellite measurements of dust in the AMA region makes the validation of this result difficult. Our modelled burdens of dust in the ATAL are larger than what has been reported in the past (e.g. Fadnavis et al., 2013; Yu et al., 2015; Fairlie et al., 2020). The higher amount of dust found in our model could be due to excessive convective transport, a lack of secondary activation of aerosols entrained into convective updrafts, a too strong dust transport in the upper troposphere from Africa and the Middle East (Wu et al., 2019), as well as the sensitivity of dust emissions to the resolution of the model (Brühl et al., 2018; Wu et al., 2019).

The differences between the simulated dust burdens between different models can be linked to the different physical processes computed for dust emissions (e.g. wind speed, hydrological parameters and soil properties).

Apart from dust, the average partitioning for other aerosol types contained in the ATAL (from anthropogenic and from biomass burning emissions) is the following: 40% Sulfate, 30% Secondary Organic Aerosols, 15% Primary Organic Matter, 14% Ammonium and less than 3% Black Carbon. Nitrate aerosols are expected to be an important aerosol component in Asia (e.g. Höpfner et al., 2019) due to the increase of nitrogen-oxides and ammonia emissions, but are not simulated in our work

For non-dust aerosols the accumulation mode dominates the anthropogenic and biomass burning ATAL aerosols. A marked positive trend of anthropogenic and biomass burning aerosol concentrations is found, with up to a factor of two increase of mass concentrations between 2000 and 2015. It is important to note that the simulated aerosol trends depend on the emissions inventory used. For example, Zheng et al. (2018) have shown that after 2013 the $SO_2$ China's emissions have decreased due to the implementation of desulfurization systems in power plants. However, this recent inventory is not included in the CEDS emission inventory used in this work and this could have some different implications in the trends we have calculated.

Our simulations reveal a double-peak structure in the vertical profile of aerosols of the ATAL, highlighting the contribution of two types of aerosols, i.e. 'cloud-borne' aerosols, including those from convective clouds and 'clear-sky/dry' aerosols. The CESM-MAM7 simulations have allowed us to analyze separately the contributions of these two types of aerosols. Dry aerosols contribute to the higher peak (peaking around 80-120 hPa) and convective cloud-borne aerosol

to the lower peak (peaking around 200-250 hPa). We show that the contribution of the convective cloud-borne aerosols to the ATAL generally increases during the phases of mature and late ATAL, in July-August, shifting the maximum of aerosol concentrations to lower altitudes. The dry aerosols are generally dominating in the early phases of the ATAL. This "double-peak" vertical structure has been observed in recent balloon and aircraft campaigns (e.g. Vernier et al., 2018; Höpfner et al., 2019) but has not been discussed in detail so far. These observations support our simulation results, which in turn provide a possible explanation for the observations. Given the uncertainties discussed throughout the paper, the ability of our simulations to represent the reality of the convective transport in the ASM is not entirely clear but the model results provide hypotheses for follow-up studies.

The obtained AOD values show an enhancement by a factor ~1.5-2.0 between the 200-80 hPa and 120-80 hPa levels. Relatively large AOD values are observed for the 200-80 hPa layer increasing from 0.007 in 2000 to 0.016 in 2015. These large values mirror the fact that extinction coefficients take into account the complete double-peak ATAL, including both dry and convective cloud-borne aerosols.

*Acknowledgments*

The authors wish to thank the CaSciModOT structure (Calcul Scientifique et Modélisation Orléans-Tours), part of the French national network of complex systems (RNSC - Réseau National des Systèmes Complexes), thanks to which the simulations could be completed.

The authors are thankful for the financial support of ANR (Agence Nationale de La Recherche) under grant ANR-17-CE01-0015 (TTL-Xing). Support from the VOLTAIRE project (ANR-10-LABX-100-01) funded by ANR through the PIA (Programme d'Investissement d'Avenir) is gratefully acknowledged. CK was funded by Deutsche Forschungsgemeinschaft (DFG, German Research Foundation) - 409585735.

AB also would like to thank the NCAR/CESM online discussion board for many helpful technical discussions that helped throughout this study, specially thanks to Louisa Emmons and Simone Tilmes.

Furthermore, the authors thank the ACE-FTS and MLS teams.

*Data availability*

MERRA-2 reanalysis data are available at http://rda.ucar.edu/datasets/ds313.3/

CMIP6 emissions files are available at https://svn-ccsm-inputdata.cgd.ucar.edu/trunk/inputdata/atm/cam/chem/emis/CMIP6_emissions_1750_2015/

ACE-FTS https://databace.scisat.ca/level2/

MLS data https://mls.jpl.nasa.gov/data/

*Code availability*

The release version 1.2.2 of CESM can be download from http://www.cesm.ucar.edu/models/cesm1.2/tags/index.html#CESM1_2_2

*Author Contribution*

AB, PS, GB and FJ designed the research and the analyse and interpretation of the model results. AB performed the model simulation with the support from FJ. CK performed the satellite analysis from MLS and ACE-FTS data. BL was involved in the discussion and results interpretation. AB prepared the manuscript with the contribution and discussions from all the co-authors.

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
