# Peer review of "Global modelling studies of composition and decadal trends of the Asian Tropopause Aerosol Layer"

_Atmospheric Chemistry and Physics, 2020_

## Referee Comment (RC1) · Anonymous Referee #1 · 1 Sep 2020

In this manuscript, Bossolasco et al. present global model investigations on the composition and evolution of the aerosol layer present in the upper troposphere in the region of the Asian summer monsoon, the so-called Asian Tropopause Aerosol Layer (ATAL). The identification of two separate layers with different origin of aerosols has to my knowledge not been described before. Further, the investigation of long-term trends provides new insights into the possible variability and anthropogenic influence. It would, however, be helpful to discuss and evaluate the finding of mineral dust and sulfate aerosol particles as the major constituents of the ATAL, more thoroughly in light of recent modelling studies and observational results. Provided that the detailed comments below are considered properly, I strongly suggest the paper for publication in

ACP.

Specific comments:

L32:

You may add a short not that nitrate aerosols have not been considered here. In my opinion this would help the reader from the beginning and does not at all diminish the value of the investigation.

L58, 'while it was not observed prior to that year':

In this context it should be mentioned that an ammonium nitrate aerosol layer has been observed already in 1997 (Fig., 1 in Höpfner et al., 2019).

L96, 'dust is one of the predominant aerosol over the Tibetan Plateau':

Please add the information that it has been detected up to 10 km altitude, otherwise one could be mislead to think that it has been observed at altitudes of the ATAL.

L124, chapter '2.1-The CESM-MAM7 model':

Could you add a paragraph how wet scavenging of gases, e.g. SO2 and NH3, and aerosols is handled in the model? As e.g. shown in Fairlie et al. (2020), this might be important for the modelling of sulfate in the ATAL.

L310, 'These results agree with some previous modelling studies (e.g. Fadnavis et al., 2013, Ma et al., 2019)'

I miss a bit more quantitative discussion about the degree of agreement between the actual study and the most recent ones. E.g. add also in the discussion the results by Fairlie et al. (2020).

Figure 2:

Do the units 'ng/m3' refer to STP (as e.g. in Fairlie et al., 2020, Fig. 3) or are these absolute values at the given pressure levels?

L543, chapter '4.4 - Aerosol Optical Depth (AOD) of the ATAL':

To be able to compare not only the absolute values but also the year-to-year variability (a strength of the actual study), I would strongly suggest to present a plot vs. time, like in Vernier et al. (2015), Fig. 6. This would allow a discussion model vs. measurements being more independent from the absolute values of AOD.

L555, 'It is important to mention that Vernier et al. (2015) have used hypotheses based on LiDAR observations and hypotheses on the LiDAR ratio value to derive the extinction coefficient to estimate the AOD.'

Vernier et al. (2015) have also used a depolarization filter ('cloudy pixels in the upper troposphere are removed using a volume depolarization ratio threshold of 5%') – could you discuss the possibility that due to this filter, also signals from dust may have been dismissed from the observations and what this would mean for your comparison?

L588, 'The results show that dust is the dominating aerosol type in terms of mass in the ATAL in agreement with other studies (e.g. Ma et al., 2019).'

This conclusion is too absolute in this context. I miss here a bit more balanced discussion with respect to other model results (1) and with respect to observations (2).

(1) Other models, like Fairlie et al. (2020) or Yu et al. (2015), do not predict dust as the dominating type of ATAL aerosols. E.g. Ma et al. (2019) refer to Brühl et al. (2018) who 'showed high sensitivity of mineral dust reaching the UTLS to model resolution, owing mostly to the differences in convection top height and overshooting convection in the parameterizations'. Could you discuss your results with respect to possible reasons why these differences between models occur? Can you detect a single cause why your model results indicate a stronger contribution of dust than other models?

(2) As long as measurements do not confirm the model results, one cannot conclude as firmly as done here about the composition of ATAL aerosols. E.g. in-situ airborne observations during the StratoClim campaign have neither identified dust nor sulfate

as a major constituents of the ATAL layer (e.g. Höpfner et al., 2019).

Technical comments:

L23, 'We identify a "double-peak" aerosols vertical profile':

e.g. 'vertical profile of aerosols' 'aerosols' is in this way often used incorrectly. Please check and correct throughout the manuscript.

L75, 'ammont':

'amount'

L76, 'niitrate':

'nitrate'

L83, 'principal aerosols typology':

'principal typology of aerosols'

L85, 'enhancement':

'enhanced'

L90, 'This region have been'

'...has been'

L112, 'aerosols':

'aerosol'

L390, 'aerosols':

'aerosol'

L384,387, '1.0 10-3 km-1'

Please use correct notation for ACP.

L391, 'seen .'

'seen.'

L450, 'details'

'detail'

References:

Brühl, C., Schallock, J., Klingmüller, K., Robert, C., Bingen, C., Clarisse, L., Heckel, A., North, P., and Rieger, L.: Stratospheric aerosol radiative forcing simulated by the chemistry climate model EMAC using Aerosol CCI satellite data, Atmos. Chem. Phys., 18, 12845–12857, https://doi.org/10.5194/acp-18-12845-2018, 2018.

Fairlie, T. D., Liu, H., Vernier, J.‐P., Campuzano‐Jost, P., Jimenez, J. L., Jo, D. S., Zhang, B., Natarajan, M., Avery, M. A., and Huey, G.: Estimates of Regional Source Contributions to the Asian Tropopause Aerosol Layer Using a Chemical Transport Model, J. Geophys. Res., 125, https://doi.org/10.1029/2019JD031506, 2020.

Höpfner, M., Ungermann, J., Borrmann, S., Wagner, R., Spang, R., Riese, M., Stiller, G., Appel, O., Batenburg, A. M., Bucci, S., Cairo, F., Dragoneas, A., Friedl-Vallon, F., Hünig, A., Johansson, S., Krasauskas, L., Legras, B., Leisner, T., Mahnke, C., Möhler, O., Molleker, S., Müller, R., Neubert, T., Orphal, J., Preusse, P., Rex, M., Saathoff, H., Stroh, F., Weigel, R., and Wohltmann, I.: Ammonium nitrate particles formed in upper troposphere from ground ammonia sources during Asian monsoons, Nature Geosci, 12, 608–612, https://doi.org/10.1038/s41561-019-0385-8, 2019.

Ma, J., Brühl, C., He, Q., Steil, B., Karydis, V. A., Klingmüller, K., Tost, H., Chen, B., Jin, Y., Liu, N., Xu, X., Yan, P., Zhou, X., Abdelrahman, K., Pozzer, A., and Lelieveld, J.: Modeling the aerosol chemical composition of the tropopause over the Tibetan Plateau during the Asian summer monsoon, Atmos. Chem. Phys., 19, 11587–11612, https://doi.org/10.5194/acp-19-11587-2019, 2019.

Vernier, J.-P., Fairlie, T. D., Natarajan, M., Wienhold, F. G., Bian, J., Martinsson, B. G., Crumeyrolle, S., Thomason, L. W., and Bedka, K. M.: Increase in upper tropospheric and lower stratospheric aerosol levels and its potential connection with Asian pollution, Journal of geophysical research. Atmospheres JGR, 120, 1608–1619, https://doi.org/10.1002/2014JD022372, 2015.

Yu, P., Toon, O. B., Neely, R. R., Martinsson, B. G., and Brenninkmeijer, C. A. M.: Composition and physical properties of the Asian Tropopause Aerosol Layer and the North American Tropospheric Aerosol Layer, Geophys. Res. Lett., 42, 2540–2546, https://doi.org/10.1002/2015GL063181, 2015.

---

## Referee Comment (RC2) · Anonymous Referee #2 · 8 Sep 2020

Comments on Manuscript No. ACP-2020-677

In the last decade the Asian Tropopause Aerosol Layer (ATAL) becomes in the focus of attention. However, the current knowledge of the ATAL is limited. This study presents model results based on the Community Earth System Model (CESM 1.2) with the focus to simulate the chemical composition of the ATAL and its decadal trends. A vertical 'double-peak' structure is found for the ATAL. Mineral dust is the dominant aerosol by mass in the ATAL. Further, the ATAL is composed of around 40 % of sulfate, 30% of secondary and 15% of primary organic aerosols, 14% of ammonium aerosols and less than 3% of black carbon. A positive trend for all aerosols was simulated using the

[Figure]

Modal Aerosol Model MAM7.

Despite of the somewhat weak discussion of the scientific results compared to the current scientific knowledge, that could be improved, this is an interesting study, which merits its publication in ACP. However, I suggest some revisions to make this possible.

1) The vertical 'double-peak' structure in the ATAL presented in this paper is very interesting. It would be an added value for the paper to discuss whether also measured vertical ATAL profiles from in situ balloon or aircraft measurements show such a 'double-peak' structure (e.g. published by Vernier et al, 2018; Brunamonti et al, 2018; Höpfner et al, 2019;...). Such a discussion could be presented in a separate 'discussion section'.

All these references are already in the publication list.

2) At several places within the paper, I am missing the discussion about what is new or different to previous publications (see specific comments below).

3) The comparison of CO between model and satellite measurements is somewhat weak in particular the comparison of vertical CO profiles (see specific comment below).

Specific comments:

P1/L19: 'pollutants' -> 'ATAL aerosols and their chemical precursors'

P1/L19: Please clarify: 'its atmospheric chemical processes'

P1/L19: What about the variabiliy of the sources/emissions contributing to ATAL?

P1/L14-19: For better understanding, I recommend to separate this long sentence into two or more sentences.

P2/L27: 'We find that mineral dust is the dominant aerosol by mass in the ATAL showing a large interannual variability, but no long-term trend, due to its natural variation.' Here it is unclear if mineral dust is dominant in both ATAL peaks. Please clarify.

P2/L40: 'The upper atmospheric circulation is dominated by the related Asian Monsoon Anticyclone (AMA), which is known to contain enhanced concentration of tropospheric trace gases and aerosols, ...' Please add some references.

P2/L59: Höpfner et al, 2019 should be also mentioned. They reported that enhanced concentrations of solid ammonium nitrate particles were found in the Asian monsoon anticyclone in 1997.

P2/L74: model, Fairlie ... -> model. Fairlie ...

P2/L76: niitrate -> nitrate

P2/L80: One result of the paper is that dust is the dominant aerosol by mass in the ATAL. However, only Ma et al (2019) is discussed here in the introduction as a model study that also found that mineral dust contribute to the ATAL. In the literature there are more studies analyzing the contribution of mineral dust to ATAL (e.g. Lau et al, 2018; Fadnavis et al, 2013, ...). Please discuss here also the results of more previous publications to give them credit.

P3/L90: What about the contribution from the Sichuan Basin (China)?

P3/L93: 'Continental convective regions have also been shown to be the main contributors to the air trapped within the AMA with North India and South of the Tibetan Plateau as specific source areas (Tissier and Legras, 2016; Legras et al., 2019).' -> (e.g. Tissier and Legras, 2016; Legras et al., 2019).' There are several other studies related to the possible source regions of AMA. Please discuss a few more of these studies. Moreover, Fairlie et al. (2020) indicated the dominance of the contribution of regional anthropogenic emissions from China and the Indian subcontinent to the ATAL. Therefore, please add here also studies discussing possible source regions of the ATAL (e.g. Fairlie et al., 2020;...)

P3/L94: Its confusing that the contribution of dust to the ATAL was discussed in several places within the introduction (see comment above).

[Figure]

P3/L108: 'Wei et al., (2019) have also found that the AMA exhibits intraseasonal variability between the Iranian Plateau and the Tibetan Plateau with a quasy-biweekly oscillation.' The bimodal distribution of the AMA is already discussed in previous publications (e.g. Zhang et al., 2002; Yan et al., 2011; Vogel et al., 2015; Nützel et al., 2016). Please add some of these references.

Zhang, Q., Wu, G., and Qian, Y.: The Bimodality of the 100 hPa South Asia High and its Relationship to the Climate Anomaly over East Asia in Summer, J. Meteorol. Soc. Jpn., 80, 733–744, 2002.

Yan, R.-C., Bian, J.-C., and Fan, Q.-J.: The impact of the South Asia High Bimodality on the chemical composition of the upper troposphere and lower stratosphere, Atmos. Oceanic Sci. Lett., 4, 229–234, 2011.

Nützel, M., Dameris, M., and Garny, H.: Movement, drivers and bimodality of the South Asian High, Atmos. Chem. Phys., 16, 14 755–14 774, https://doi.org/10.5194/acp-16-14755-2016, 2016.

P7/L256: 'This could explain the low bias in CO mixing ratios for our comparisons with satellite measurements.' What about the impact of vertical transport from surface sources to the UTLS in the model. In Fig. 1e the CO values at around 500hPa are underestimated in the model, however above ∼150hPa model and measurement agree. However the CO increase between 500 and 150hPa in the measurements is much lower compared to the model. Could that be a hint on vertical transport issues in the model?

P7/L262: In the literature 'eddy shedding' is not the same as the bimodality of the AMA. Please clarify.

P7/L265: 'They show a distributed pattern with maxima above eastern Asia, but also above western Asia (Fig. 1d), ...'. What about the maxima over Africa near the Equator?

P8/L274: 'We have also tested the vertical structures of CESM-MAM7 simulations, using an ACE-FTS CO mixing ratio profile in the UTLS (Fig. 1e).' One single vertical CO profile is not very representative for a simulation over 16 years. Please could you provide more vertical CO profiles and maybe present their mean value and its variability over an larger time frame perhaps for June, July and August (similar as Fig. 3). Is the vertical 'double-peak' found in aerosol also present in simulated CO?

P10/L322: '(Randel and Park, 2006; Garny...)'-> '(e.g. Randel and Park, 2006; Garny...)'

P12/L359: 'The vertical structure of the AMA-related dynamics has been investigated by several authors (Bergman, J. et al., 2013; Garny and Randel, 2013; Brunamonti et al., 2018)..' Remove 'J' after Bergman and add 'e.g.' . There are more previous publications studying the vertical structure of the AMA (e.g. Park et al. 2009; Vogel et al, 2019; Bian et al, 2020,..)

Park M, Randel WJ and Emmons LK et al. Transport pathways of carbon monoxide in the Asian summer monsoon diagnosed from Model of Ozone and Related Tracers (MOZART). J Geophys Res 2009; 114: D08303.

Vogel, B., Müller, R., Günther, G., Spang, R., Hanumanthu, S., Li, D., Riese, M., and Stiller, G. P.: Lagrangian simulations of the transport of young air masses to the top of the Asian monsoon anticyclone and into the tropical pipe, Atmos. Chem. Phys., 19, 6007–6034, https://doi.org/10.5194/acp-19-6007-2019, https://www.atmos-chem-phys.net/19/6007/2019/, 2019.

Bian, J., Li, D., Bai, Z., Li, Q., Lyu, D., and Zhou, X.: Transport of Asian surface pollutants to the global stratosphere from the Tibetan Plateau region during the Asian summer monsoon, Natl. Sci. Rev., 7, 516–533, https://doi.org/10.1093/nsr/nwaa005, https://doi.org/10. 1093/nsr/nwaa005, 2020.

P14/Fig.3: Please explain briefly in the Figure caption why an application of an extinction filter is shown.

P18/L485: Please explain the meaning of the p-value in words.

P18/L506: This mirrors the increase of the emissions in Asia.' Zheng et al. (2018) shows that after 2013 China's anthropogenic emission of some pollutants decreased substantially (e.g., $SO_2$) because of the implementation of new emission control measures. How does that fit to your results about increasing emissions in Asia? Are the new Chinese emission control measures considered in the Regional Emission inventory that is used in this study?

Zheng, B., Tong, D., Li, M., Liu, F., Hong, C., Geng, G., Li, H., Li, X., Peng, L., Qi, J., Yan, L., Zhang, Y., Zhao, H., Zheng, Y., He, K., and Zhang, Q.: Trends in China's anthropogenic emissions since 2010 as the consequence of clean air actions, Atmos. Chem. Phys., 18, 14095–14 111, https://doi.org/10.5194/acp-18-14095-2018, https://www.atmos-chem-phys.net/18/14095/2018/, 2018.

p17/Fig.4: Figure 4 shows very nicely the impact of volcanic eruptions. Is there also a modulation by El Niño?

P18/L511: Please clarify the meaning of 'increment' and 'correlation'.

P21/L559: Please clarify 'Our double-peak ATAL features highlighted in Fig. 6a'. I assume the meaning is that two maxima of AOD at different longitudes are found (corresponding to the bimodality of the AMA). It is confusing here because the expression 'double-peak structure' was already used for the two maxima found in the vertical structure of the ATAL. Or is there an misunderstanding? Please clarify.

P21/L564: 'The difference between the AOD values obtained for the two altitude ranges in Fig 6a and 6b points at the importance of what we have identified as convective incloud aerosols.' Please explain this in more detail.

P21/L567: 'full double-peak ATAL' (see above L559)

P22/L585: 'The model evaluation with MLS and ACE-FTS satellite data reflects that transport and convection features are well represented in our simulations, despite a possible underestimation of the biomass burning emissions.' In the paper, a rough comparison between simulated CO and measured CO is shown. I would not call this 'model evaluation'. Further, I am not sure if the transport and convection features are overall well represented in the model (see comment to L256). Please rephrase this sentence and use a somewhat more cautious formulation.

P22/L590: '..what has been reported in the past'. Please add some references.

P22/L595: 'Apart from dust, the average partitioning for other aerosol types contained in the ATAL (from anthropogenic and from biomass burning emissions) is the following: 40% Sulfate, 30% Secondary Organic Aerosols, 15% Primary Organic Matter, 14% Ammonium and less than 3% Black Carbon.' What is new or different compared to previous results regarding the chemical composition of ATAL?

P22/L602: '... a marked positive trend of anthropogenic and biomass burning aerosol concentrations is found, with up to a factor two increase of mass concentrations between 2000 and 2015. ' What are the consequences if the ATAL over Asia is increasing further in future?

---

## Referee Comment (RC3) · Anonymous Referee #3 · 14 Sep 2020

**General comments:**

This study is very well written and addresses a hot topic in the scope of ACP. It provides interesting hypotheses about the nature of the ATAL, e.g. that there is a double-peak vertical structure, mineral dust dominates aerosol mass, and that the ATAL signature has been increasing from 2000 to 2015. This should be published, considering the following.

The paper would benefit from actually working out if one or more of the above hypotheses have something to do with reality. In the current version, the analyses and discussions are limited almost exclusively to the modelling world, to one simulation.

[Figure]

This simulation is linked to the real world just by a comparison to observed CO. However, the emissions contributing to ATAL have other source distributions than CO, and are affected by other processes.

Furthermore, the uplift of air from the ground to the UTLS - a crucial process for ATAL – might need a closer look in the model: simulated CO compares favourably to the observations in the UTLS, despite being off in the free troposphere (Figure 1e). A much more thorough model evaluation would be appropriate, covering (proxies for) all species, precursors and processes of relevance for the aspects of ATAL that are discussed in this study. Sensibly splitting this between the supplement and the main text would allow the paper to stay concise.

Apart from the mere model evaluation, it would help putting some effort into researching available observations for support of the model-based findings about ATAL.

A more detailed understanding of the strengths and weaknesses of the simulation might also help the discussion of how this study compares to other modelling results.

**Specific comments:**

L87: have -> has

L105: The bimodality of the AMA has been discussed for longer, see e.g. (Nützel et al. 2016, Pan et al. 2016) and references therein.

L107: beweekly -> biweekly or bi-weekly

L116: larger aerosols composition -> more comprehensive aerosol composition

L116: Please check the use of aerosols / aerosol / aerosols' / . . . throughout the paper.

L149: Isn't anvil associated with convective rather than stratiform clouds?

L179: Simulated ATAL trends are likely to critically depend these assumptions. Please

elaborate on the uncertainties in the emissions' setup, providing the reader with a sense on how this might print through to the results for ATAL.

L191: Different reanalyses have different peculiarities in representing AMA (see e.g. Nützel et al. 2016). Please shortly note whether there is something specific the reader needs to know about MERRA.

L281: A 30

L303: Please consider showing the comparison to the corresponding observations.

L330: Is the following understanding correct? There is no dynamic tracking of the AMA. Rather you choose a static box, which most of the time is part of the AMA. Any averages should thus be dominated by AMA conditions. This is ok, but some rewording might help to make the approach clearer.

L335: Isentropic surfaces might be better to describe horizontal transport and thus the horizontal extent of the ATAL (Santee et al. 2017, Gottschaldt et al. 2018). Please check whether or not your results crucially depend on the choice of the vertical coordinate system.

L363: The term "mode" is already in use for aerosol size ranges and for the dynamics of the AMA. Does it refer to different aerosol classes here?

L363: Is there any observational hint for such a double-peak layering?

L397: "Double-peak", when used as adjective? Please check throughout the paper.

L405: Please mention in the caption that this is modelling only.

L433: That is rather vague. Several models get an ATAL, so it seems to be a quite stable feature. Interestingly, the exact characteristics vary, probably depending on the various factors listed here. For improving our understanding of ATAL it is therefore important to really understand the model differences, and to find those explanations that are supported by observations.

[Figure]

L457: Please use subscripts in chemical compounds throughout the paper.

L484: showed -> shown

L493: Is there a chance to be more specific: Which aspects of the dust cycle are captured well by your model? Which are not and what are the implications for your conclusions about ATAL?

L503: Here you state model shows that increased emissions translate into enhanced ATAL, but in L567 alternative explanations are offered. Please check consistency. Furthermore, as already noted by reviewer2, emission trends are more complicated. A more detailed analysis might be needed, e.g. explicitly correlate emissions (by region) with ATAL parameters.

L552: This formulation is kind of suggesting that Vernier et al. might be wrong. Please elaborate.

L555: Is CESM1/CARMA from the same model family you are using? Then getting similar results could also indicate a common problem.

L562: Another interesting hypothesis. Please check whether there are any observations supporting it.

L585: Please consider rewording: The results show ... -> Our modelling results indicate ...

L594: Please make it clear from the beginning that nitrate aerosols might be an important aspect you omit.

L812: space between references missing

**References:**

Barret, B., B. Sauvage, Y. Bennouna E. Le Flochmoen (2016) Upper-tropospheric

CO and O3 budget during the Asian summer monsoon. Atmospheric Chemistry and Physics, 16, 9129-9147.

Gottschaldt, K.-D., H. Schlager, R. Baumann, D. S. Cai, V. Eyring, P. Graf, V. Grewe, P. Jöckel, T. Jurkat-Witschas, C. Voigt, A. Zahn  H. Ziereis (2018) Dynamics and composition of the Asian summer monsoon anticyclone. Atmospheric Chemistry and Physics, 18, 5655-5675.

Ma, J., C. Brühl, Q. He, B. Steil, V. A. Karydis, K. Klingmüller, H. Tost, B. Chen, Y. Jin, N. Liu, X. Xu, P. Yan, X. Zhou, K. Abdelrahman, A. Pozzer  J. Lelieveld (2019) Modeling the aerosol chemical composition of the tropopause over the Tibetan Plateau during the Asian summer monsoon. Atmos. Chem. Phys., 19, 11587–11612.

Nützel, M., M. Dameris  H. Garny (2016) Movement, drivers and bimodality of the South Asian High. Atmospheric Chemistry and Physics, 16, 14755-14774.

Pan, L. L., S. B. Honomichl, D. Kinnison, M. Abalos, W. J. Randel, J. W. Bergman J. Bian (2016) Transport of chemical tracers from the boundary layer to stratosphere associated with the dynamics of the Asian summer monsoon. Journal of Geophysical Research: Atmospheres, 121, 1-16.

Santee, M. L., G. L. Manney, N. J. Livesey, M. J. Schwartz, J. L. Neu  W. G. Read (2017) A comprehensive overview of the climatological composition of the Asian summer monsoon anticyclone based on 10 years of Aura Microwave Limb Sounder measurements. Journal of Geophysical Research: Atmospheres, 122, 5491-5514.

---

## Author Comment (AC1) · 12 Nov 2020

We would like to thank Reviewer 1 for her/his comments and his/her fast response during the discussion phase, which helped us improve the quality of the manuscript. We have discussed the suggestions/corrections raised by Reviewer 1 with the coauthors and made the changes in the text accordingly. Below each comment you can find our answers and the respective changes made in the manuscript.

Note:
Due to changes made in the manuscript, some of the line numbers referred by the reviewers have changed. These changes are shown in green when applicable.

**Anonymous Referee #1**

In this manuscript, Bossolasco et al. present global model investigations on the composition and evolution of the aerosol layer present in the upper troposphere in the region of the Asian summer monsoon, the so-called Asian Tropopause Aerosol Layer (ATAL). The identification of two separate layers with different origin of aerosols has to my knowledge not been described before. Further, the investigation of long-term trends provides new insights into the possible variability and anthropogenic influence.
It would, however, be helpful to discuss and evaluate the finding of mineral dust and sulfate aerosol particles as the major constituents of the ATAL, more thoroughly in light of recent modelling studies and observational results. Provided that the detailed comments below are considered properly, I strongly suggest the paper for publication in ACP

Specific comments:
1) L32:
You may add a short not that nitrate aerosols have not been considered here. In my opinion this would help the reader from the beginning and does not at all diminish the value of the investigation.

**Done**

2) L58, 'while it was not observed prior to that year':
In this context it should be mentioned that an ammonium nitrate aerosol layer has been observed already in 1997 (Fig., 1 in Höpfner et al., 2019).

**Done. A small sentence about this was added at the end of the paragraph.**

3)  L96, 'dust is one of the predominant aerosol over the Tibetan Plateau':

Please add the information that it has been detected up to 10 km altitude, otherwise one could be mislead to think that it has been observed at altitudes of the ATAL.

**The paragraph was changed accordingly, as follow:**
"In several studies, dust has been shown as a major contributor to the aerosol burden in the Asian upper troposphere during summer. Xu et al. (2015), using CALIOP and MISR (Multi-angle Imaging SpectroRadiometer) satellite data, **have found that dust is one of the predominant aerosol over the Tibetan Plateau most probably originating from the Taklamakan desert and lofted from the surface to an altitude of about 10 km**."

4) L124/L145 (new line number), chapter '2.1-The CESM-MAM7 model':
Could you add a paragraph how wet scavenging of gases, e.g. SO2 and NH3, and aerosols is handled in the model? As e.g. shown in Fairlie et al. (2020), this might be important for the modelling of sulfate in the ATAL.

**We have added two paragraphs to explain shortly the wet removal of soluble gases and aerosols (L168-176 and 200-207, in the revised manuscript). The wet scavenging used in CESM-MAM7 is the standard scheme in CAM5, although as has been noted by Fairlie et al. (2020) a more physically based treatment of wet scavenging of SO$_2$ in convective updrafts increases the amount of sulfate. A more detailed study to evaluate this will be done in a future.**

5) L310/L388 (new line number), 'These results agree with some previous modelling studies (e.g. Fadnavis et al.,2013, Ma et al., 2019)'
I miss a bit more quantitative discussion about the degree of agreement between the actual study and the most recent ones. E.g. add also in the discussion the results by Fairlie et al. (2020).

**We thank the Reviewer for this correction. We have extended the discussion and added a more detailed discussion about the degree of agreementbetween our study and the recent ones and the possible biases of dust modelled. See the lines 388-419 in the revised manuscript.**

6) Figure 2:
Do the units 'ng/m3' refer to STP (as e.g. in Fairlie et al., 2020, Fig. 3) or are these absolute values at the given pressure levels?

**They are absolute values. These units have been used to quantify aerosol burdens in several previous studies (e.g. Fadnavis et al., 2019; Fadnavis et al., 2017; Ma et al., 2019). We use them for the sake of comparison with these previous studies; we acknowledge that other authors use other units, as volume mixing ratios.**

*Fadnavis, S., Müller, R., Kalita, G., Rowlinson, M., Rap, A., Li, J.-L. F., Gasparini, B. and Laakso, A.: The impact of recent changes in Asian anthropogenic emissions of SO2 on sulfate loading in the upper troposphere and lower stratosphere and the associated radiative changes, Atmos. Chem. Phys., 19(15), 9989–10008, doi:10.5194/acp-19-9989-2019, 2019.*

7) L543/L653 (new line number), chapter '4.4 - Aerosol Optical Depth (AOD) of the ATAL':
To be able to compare not only the absolute values but also the year-to-year variability (a strength of the actual study), I would strongly suggest to present a plot vs. time, like in Vernier et al. (2015), Fig. 6. This would allow a discussion model vs. Measurements being more independent from the absolute values of AOD.

**In the revised manuscript, we have now provided such plots and a corresponding discussion. Here we summarize that discussion. On the plot in Fig. 6 in the revised manuscript: the associated summer-averaged AOD values are larger than those of Vernier et al. (2015) (their Fig. 6). It is difficult to directly and quantitatively compare our simulated AODs with the measurements of Vernier et al. 2015 because of the different considered periods and, more important, the cloud filtering in the AOD observations (see next answer). First, periods impacted by volcanic aerosol perturbations are somewhat different between our model analysis and Vernier et al. (2015) (e.g. we have excluded years 2006-2008). We are confident that our selection of volcanic-free periods is state-of-the-art (see manuscript for details). Second, we may expect that the cloud screening result in different average AODs and trends for simulations and observations. This is an obvious reason why we have found a lower altitude/in-cloud ATAL component (lower-altitude peak in the double-peak structure), which is not observed in Vernier et al. 2015. Our new AOD time-series shows that accounting for the double-peak ATAL structure leads to different trends and reflects the importance of the altitude range used to estimate the year-to-year variability. We then conclude that both studies reveal an increasing ATAL AOD trend but without directly reconciling the two datasets, as a result of different methods applied.**

8) L555/L663 (new line number), 'It is important to mention that Vernier et al. (2015) have used hypotheses based on LiDAR observations and hypotheses on the LiDAR ratio value to derive the extinction coefficient to estimate the AOD.'
Vernier et al. (2015) have also used a depolarization filter ('cloudy pixels in the upper troposphere are removed using a volume depolarization ratio threshold of 5%') – could you discuss the possibility that due to this filter, also signals from dust may have been dismissed from the observations and what this would mean for your comparison?

**The paragraph was changed accordingly, and we discuss irregularly-shaped particles might have been removed due to the depolarization filter applied by Vernier et al. 2015, with a possible impact on dust.**

9) L588/L712 (new line number), 'The results show that dust is the dominating aerosol type in terms of mass in the ATAL in agreement with other studies (e.g. Ma et al., 2019).'
This conclusion is too absolute in this context. I miss here a bit more balanced discussion with respect to other model results (1) and with respect to observations (2).

(1) Other models, like Fairlie et al. (2020) or Yu et al. (2015), do not predict dust as the dominating type of ATAL aerosols. E.g. Ma et al. (2019) refer to Brühl et al. (2018) who 'showed high sensitivity of mineral dust reaching the UTLS to model resolution, owing mostly to the differences in convection top height and overshooting convection in the parameterizations'. Could you discuss your results with respect to possible reasons
why these differences between models occur? Can you detect a single cause why your model results indicate a stronger contribution of dust than other models?

**The paragraph has been changed and a more detailed discussion has been included in Sect. 4.1 (see answer 5) and in the conclusion. A brief summary of the new section is in the following. We are aware of the larger amounts of dust in the UTLS region, in our simulation, with respect to some previous works. Nevertheless, dust modelling is still uncertain, at the present time. As discussed in previous works and especially in Wu et al. 2019, who compare different dust schemes with satellite observations, the GCMs have large uncertainties in the simulated dust cycle in terms of spatial distribution and temporal variation. Possible reasons, which would require thorough analyses, could be linked to uncertainties in the physical process leading to dust erosion, the representation of convection in the models and/or effects of vertical resolution on the transport. The simulations of Brühl et al. (2018) have shown that the amounts of dust reaching the UTLS region in the EMAC model are sensitive to model resolution. In our work with CESM-MAM7, we use a 1.9 x 2.5° horizontal resolution and 56 vertical levels which is standard for CESM1 and has been used in previous studies of aerosol properties (Yu et al. 2015, 2017). These resolutions are lower than the values in Brühl et al. (2018) and this could indeed impact the dust reaching the UTLS in CAM5 as result of differences in convection top height and overshooting convection.**
**We use also use the standard configuration of CAM5 for the vertical transport (Zhang and McFarlane, 1995). This information has been added in Sec. 2 (L243-249).**
**We feel that detailed analyses on resolution and convective parameterization would be largely out from the scope of the paper. It is worth noticing that large uncertainties exist also in satellite retrievals of dust, which makes difficult the validation of the models.**

*Zhang, G. J., and N. A. McFarlane (1995). Sensitivity of climate simulations to the parameterization of cumulus convection in the Canadian Climate Centre general circulation model, Atmos. Ocean, 33, 407–446.*

10) (2) As long as measurements do not confirm the model results, one cannot conclude as firmly as done here about the composition of ATAL aerosols. E.g. in-situ airborne observations during the StratoClim campaign have neither identified dust nor sulfate as a major constituents of the ATAL layer (e.g. Höpfner et al., 2019).

**We agree and we have toned down some statements and discussion throughout the manuscript, including the "validation" of the model. However due the lack of information about the ATAL composition derived from observations in the AMA region over the period of the simulation we can only qualitatively compare our model results with the ion chromatography analysis from aerosol samples collected in summer 2015 in India during the BATAL balloon campaign (Vernier et al., 2018). As discussed in Vernier et al 2018, the undetectable concentration of sulfate ions (<10 ng.m$^{-3}$) seems to be contradictory with the expectation of a major contribution of sulfur (and influence emissions in Asia over the past few decades) in the aerosol layer in the UTLS. This result strongly differs from the observations of the StratoClim campaign which has identified a very high proportion of nitrates and sulfates in the summer 2017 ATAL, reflecting the complexity of the processes controlling the ATAL composition and variability. The ATAL might be so variable that a single campaign is not sufficient to characterise it. This is reflected by the revised text in the new manuscript version.**

Technical comments:

11) L23, 'We identify a "double-peak" aerosols vertical profile':
e.g. 'vertical profile of aerosols' 'aerosols' is in this way often used incorrectly. Please check and correct throughout the manuscript.
**Changed**

12) L75/L88 (new line number), 'ammont':
'amount'
**Corrected**

13) L76/L89 (new line number), 'niitrate':
'nitrate'
**Corrected**

14) L83/L85 (new line number), 'principal aerosols typology':
'principal typology of aerosols'
**Corrected**

15) L85/L91 (new line number), 'enhancement':
'enhanced'
**Corrected**

16) L90/L108 (new line number), 'This region have been'
'. . .has been'
**Changed. We have deleted the paragraph because it was redundant and it was extended in accordance with comments of reviewer #2.**

17) L112/L133 (new line number), 'aerosols':
'aerosol'
**Corrected**

18) L390/L514 (new line number), 'aerosols':

'aerosol'
**Corrected**

19) L384,387/ L508,511 (new line number), '1.0 10-3 km-1'
Please use correct notation for ACP.
**Corrected**

20) L391/L515 (new line number), 'seen .'
'seen.'
**Corrected**

21) L450/L568 (new line number), 'details'
'detail'
**Corrected**

---

## Author Comment (AC2) · 12 Nov 2020

**We would like to thank Reviewer 2 for the time spent and the detailed comments and suggestions (including additional literature). This helped us improving our manuscript. In the following, we address each comment individually, including the changes we made to the manuscript accordingly.**

Note:
Due to changes made in the manuscript, some of the line numbers referred by the reviewers have changed. These changes are shown in green when applicable.

**Anonymous Referee #2**

Comments on Manuscript No. ACP-2020-677
In the last decade the Asian Tropopause Aerosol Layer (ATAL) becomes in the focus of attention. However, the current knowledge of the ATAL is limited. This study presents model results based on the Community Earth System Model (CESM 1.2) with the focus to simulate the chemical composition of the ATAL and its decadal trends. A vertical 'double-peak' structure is found for the ATAL. Mineral dust is the dominant aerosol by mass in the ATAL. Further, the ATAL is composed of around 40 % of sulfate, 30% of secondary and 15% of primary organic aerosols, 14% of ammonium aerosols and less than 3% of black carbon. A positive trend for all aerosols was simulated using the Modal Aerosol Model MAM7.

Despite of the somewhat weak discussion of the scientific results compared to the current scientific knowledge, that could be improved, this is an interesting study, which merits its publication in ACP. However, I suggest some revisions to make this possible.

1) The vertical 'double-peak' structure in the ATAL presented in this paper is very interesting. It would be an added value for the paper to discuss whether also measured vertical ATAL profiles from in situ balloon or aircraft measurements show such a 'double-peak' structure (e.g. published by Vernier et al, 2018; Brunamonti et al, 2018; Höpfner et al, 2019;...). Such a discussion could be presented in a separate 'discussion section'.
All these references are already in the publication list.

**A discussion regarding the "double-peak" structure observations has been added in the Sec 4.2. See lines 489-503 in the revised manuscript.**

2) At several places within the paper, I am missing the discussion about what is new or different to previous publications (see specific comments below).

**We have added more discussion about how our results compare with previous studies (Fadnavis et al., 2013; Yu et al., 2015; Ma et al., 2019, Fairlie et al., 2020).**
**The new contribution of our work is the ATAL aerosol modeled trends over 16 years (not shown before) and the new ATAL double-peak structure, which, although it has been observed in some recent measurements has not been extensively discussed. We modified the text in different places so to make it clear these two major novel results of our work.**

3) The comparison of CO between model and satellite measurements is somewhat weak in particular the comparison of vertical CO profiles (see specific comment below).
**We address the answer in the specific comments below.**

Specific comments:
4) P1/L19: 'pollutants' -> 'ATAL aerosols and their chemical precursors'
**The paragraph was changed**

5) P1/L19: Please clarify: 'its atmospheric chemical processes'
**The paragraph was changed**

6) P1/L19: What about the variabiliy of the sources/emissions contributing to ATAL?
**The reviewer is right, we have added a small sentence to mention it.**

7) P1/L14-19: For better understanding, I recommend to separate this long sentence into two or more sentences.
**The paragraph was changed.**

8) P2/L27: 'We find that mineral dust is the dominant aerosol by mass in the ATAL showing a large interannual variability, but no long-term trend, due to its natural variation.' Here it is unclear if mineral dust is dominant in both ATAL peaks. Please clarify.
**This clarification was added in parentheses.**

9) P2/L40/ L43 (new line number): 'The upper atmospheric circulation is dominated by the related Asian Monsoon Anticyclone (AMA), which is known to contain enhanced concentration of tropospheric trace gases and aerosols, ...' Please add some references.
**Done**

10) P2/L59/ L60 (new line number): Höpfner et al, 2019 should be also mentioned. They reported that enhanced concentrations of solid ammonium nitrate particles were found in the Asian monsoon anticyclone in 1997.
**This has been added**

11) P2/L74/ L86 (new line number): model, Fairlie ... -> model. Fairlie ...

**"** Using GEOS-Chem (Goddard Earth Observing System with Chemistry) chemical transport model, Fairlie et al. (2020) have found significant amont of sulfate....**"**
**We think using a comma is correct here.**

12) P2/L76 / L89 (new line number): niitrate -> nitrate
**Changed it**

13) P2/L80/ L93 (new line number): One result of the paper is that dust is the dominant aerosol by mass in the ATAL. However, only Ma et al (2019) is discussed here in the introduction as a model study that also found that mineral dust contribute to the ATAL. In the literature there are more studies analyzing the contribution of mineral dust to ATAL (e.g. Lau et al, 2018; Fadnavis et al, 2013, ...). Please discuss here also the results of more previous publications to give them credit.

**To clarify, we first mention the previous studies that are based on modelling and their results for all aerosols types in the ATAL. Then we emphasize the previous studies that showed an important contribution of dust in the ATAL region ( Lau et., al 2018, Ma et., al 2019). To further clarify the text, we have moved the paragraph regarding the discussion of dust to some lines above (Line:93-104 in the revised manuscript) and we rephrased it.**

14) P3/L90/ L115 (new line number): What about the contribution from the Sichuan Basin (China)?
**We have added a paragraph to mentioned it (including the citation to Lau et al 2018).**

15) P3/L93/ L108 (new line number): 'Continental convective regions have also been shown to be the main contributors to the air trapped within the AMA with North India and South of the Tibetan Plateau as specific source areas (Tissier and Legras, 2016; Legras et al., 2019).' -> (e.g. Tissier and Legras, 2016; Legras et al., 2019).' There are several other studies related to the possible source regions of AMA. Please discuss a few more of these studies. Moreover, Fairlie et al. (2020) indicated the dominance of the contribution of regional anthropogenic emissions from China and the Indian subcontinent to the ATAL. Therefore, please add here also studies discussing possible source regions of the ATAL (e.g. Fairlie et al., 2020;...)

**This discussion has been extended and the references added. We have modified the paragraph as follow:**
**"** Continental convective regions have also been shown to be the main contributors to the air trapped within the AMA with North India and South of the Tibetan Plateau as specific source areas (e.g. Tissier and Legras, 2016; Legras et al., 2019). Bergman et al. (2013), using Lagrangian backward trajectories, have shown that the anticyclone is connected to the boundary layer through a vertical conduit centred over Northeast India, Nepal, and southern Tibet. In the recent BATAL campaign, Vernier et al. (2018) have used back-trajectory calculations to point at North of India as a principal region source for ATAL. Lau et al. (2018), based on MERRA-2 reanalysis have reported that the Himalayas Gangetic Plain (HGP) region and the Sichuan Basin (SB) of southwestern China, are two important regions with

strong vertical transport of CO, carbonaceous aerosols and dust from the surface to the UTLS. On the other hand, the simulations of Fairlie et al. (2020)   have suggested that the anthropogenic sources from India contribute to up to 40% of sulfate and up to 65% of organic and ammonium aerosols in the western ATAL region, whereas China contributes up to 60% (both sulfate and organic aerosols) in the eastern ATAL region."

16) P3/L94: Its confusing that the contribution of dust to the ATAL was discussed in several places within the introduction (see comment above).

**We agree. The paragraph has been changed accordingly and we have moved the discussion of dust (see answer 13 above,L93).**

17) P3/L108/ L128 (new line number): 'Wei et al., (2019) have also found that the AMA exhibits intraseasonal variability between the Iranian Plateau and the Tibetan Plateau with a quasy-biweekly oscillation.' The bimodal distribution of the AMA is already discussed in previous publications (e.g. Zhang et al., 2002; Yan et al., 2011; Vogel et al., 2015; Nützel et al., 2016). Please add some of these references.
Zhang, Q., Wu, G., and Qian, Y.: The Bimodality of the 100 hPa South Asia High and its Relationship to the Climate Anomaly over East Asia in Summer, J. Meteorol. Soc.Jpn., 80, 733–744, 2002.

Yan, R.-C., Bian, J.-C., and Fan, Q.-J.: The impact of the South Asia High Bimodality on the chemical composition of the upper troposphere and lower stratosphere, Atmos. Oceanic Sci. Lett., 4, 229–234, 2011.

Nützel, M., Dameris, M., and Garny, H.: Movement, drivers and bimodality of the South Asian High, Atmos. Chem. Phys., 16, 14 755–14 774, https://doi.org/10.5194/acp-16-14755-2016, 2016.

**The reviewer is right, thanks, the paragraph has been changed as follow:**

**"** Several studies have shown that the AMA exhibits intraseasonal variability between the Iranian Plateau and the Tibetan Plateau with a quasi-biweekly oscillation( e.g. Zhang et al., 2002; Yan et al., 2011; Nützel et al., 2016; Pan et al. 2016; Wei et al., 2019).**"**

18) P7/L256 /L307 (new line number): 'This could explain the low bias in CO mixing ratios for our comparisons with satellite measurements.' What about the impact of vertical transport from surface sources to the UTLS in the model. In Fig. 1e the CO values at around 500hPa are underestimated in the model, however above ~150hPa model and measurement agree. However the CO increase between 500 and 150hPa in the measurements is much lower compared to the model. Could that be a hint on vertical transport issues in the model?

**The reasons for the differences between observed and modelled CO with respect to altitude are still unclear. We have added a**

paragraph to discuss about the possibility of this discrepencies are linked to the treatment of convection by CESM1/CAM5 (L335-347) together with discrepancies in emission inventories.

See more details below:

The model is in general able to simulate much of the large-scale behavior for CO found in space-borne observations, although the degree of consistency of simulated and observed CO amounts depends on the season and latitude of the comparison as reported with CAM4 by Park et al. (2013) (note that convection is parameterized in the same way in CAM4 and CAM5).

He et al. (2015) using CESM1/CAM5 have reported an under predictions of CO at the surface over Asia, but the global tropospheric column of CO seems to be over predicted in their study. These authors suggest uncertainties in terms of spatial allocations of CO emissions as well as convective transport treatments. The convection in CESM-MAM7 is parameterized using the Zhang-McFarlane scheme (Zhang and McFarlane, 1995) for deep convection and the Hack scheme (Hack, 1994) for shallow convection (See L243:249 in the revised manuscript). This is a typical parameterization used in numerous studies involving the CAM5 (or previous versions) model. For more details about the convection schemes used in CAM5 please see Liu et al 2012 (Supplement) http://www.geosci-model-dev.net/5/709/2012/gmd-5-709-2012-supplement.pdf.

Brühl et al. (2018) have reported that model resolution affects transport (of aerosols in their study). The model resolution used is likely to impact the calculated transport of gases by convection. In our work with CESM-MAM7, we use a 1.9 x 2.5° horizontal resolution and 56 vertical levels which is standard for CESM1 and has been used in previous studies of aerosol properties (Yu et al. 2015; Yu et al. 2017).

The model is "nudged" using external meteorological fields (here MERRA2) and although this nudging does not directly change the convection parameterization in the model, it is expected to influence the representation of convection, which is still to be properly assessed. The impacts of nudging in CCMs (including CESM1) on the vertical transport have been studied by Chrysanthou et al. (2019), who have shown some limitations in simulating the mean vertical transport in the stratosphere for these models (but interestingly with realistic representations of fast horizontal transport in their work).

Our goal is to investigate the ability of the model to simulate ATAL's properties in its typical set-up and tests about resolution and standard diagnostics of atmospheric convection in CAM 5.1 would deviate from the scope of the paper.

Finally, another possibilities could be: the uncertainties in the extrapolation emissions using CEDS inventory (this discussion has been added in the revised manuscript L308:313) and the reactivity of CO with OH which is different in the gas and liquid phase; in this case, a thorough analysis of the pertinence of simulated OH amounts and of the reaction rate of oxidation of CO by OH in presence of clouds (more predominant below the 150 hPa level

**where the difference is larger) could be conducted in a next study. In the UTLS, possible underpredictions of temperature could lead to smaller loss of CO with OH (He et al., 2015).**
**Following the reviewer's comment, we have toned down our statement that the model-observation comparisons shown in figure 1 tend to "validate" the model calculation of transport.**

*Park, M., W. J. Randel, D. E. Kinnison, L. K. Emmons, P. F. Bernath, K. A. Walker, C. D. Boone, and N. J. Livesey (2013), Hydrocarbons in the upper troposphere and lower stratosphere observed from ACE-FTS and comparisons with WACCM, J. Geophys. Res. Atmos., 118, 1964–1980, doi:10.1029/2012JD018327.*

*He, J., Y. Zhang, T. Glotfelty, R. He, R. Bennartz, J. Rausch, and K. Sartelet (2015), Decadal simulation and comprehensive evaluation of CESM/CAM5.1 with advanced chemistry, aerosol microphysics, and aerosol cloud interactions, J. Adv. Model. Earth Syst., 7, 110–141, doi:10.1002/2014MS000360.*

*Chrysanthou, Andreas, Amanda C. Maycock, Martyn P. Chipperfield, Sandip Dhomse, Hella Garny, Douglas Kinnison, Hideharu Akiyoshi, Makoto Deushi, Rolando R. Garcia, Patrick Jöckel, Oliver Kirner, Giovanni Pitari, David A. Plummer, Laura Revell, Eugene Rozanov, Andrea Stenke, Taichu Y. Tanaka, Daniele Visioni, and Yousuke Yamashita, The effect of atmospheric nudging on the stratospheric residual circulation in chemistry–climate models, Atmos. Chem. Phys., 19, 11559–11586, 2019, https://doi.org/10.5194/acp-19-11559-2019.*

19) P7/L262/ L317( new line number): In the literature 'eddy shedding' is not the same as the bimodality of the AMA. Please clarify.
**We agree, "eddy shedding" word was not used correctly in our manuscript, so we have changed in the paragraph.**

20) P7/L265/ L320 (new line muber): 'They show a distributed pattern with maxima above eastern Asia, but also above western Asia (Fig. 1d), …'. What about the maxima over Africa near the Equator?
**The CO maximum at 150 hPa over north Africa is expected to result from the Asian pollution uplifted to the upper troposphere and recirculated by the ASM as described in Barret et al. (2008).**

*B. Barret, P. Ricaud, C. Mari, J.-L. Attié, N. Bousserez, B. Josse, E. Le Flochmoën, N. J. Livesey, S. Massart, V.-H. Peuch, A. Piacentini, B. Sauvage, V. Thouret, and J.-P. Cammas, Transport pathways of CO in the African upper troposphere during the monsoon season: a study based upon the assimilation of spaceborne observations, Atmos. Chem. Phys., 8, 3231–3246, 2008.*

21) P8/L274/ L326 (new line number): 'We have also tested the vertical structures of CESM-MAM7 simulations, using an ACE-FTS CO mixing ratio profile in the UTLS (Fig. 1e).' One single vertical CO profile is not very

representative for a simulation over 16 years. Please could you provide more vertical CO profiles and maybe present their mean value and its variability over an larger time frame perhaps for June, July and August (similar as Fig. 3). Is the vertical 'double-peak' found in aerosol also present in simulated CO?

**Only a few ACE-FTS profiles are available each year in the AMA (and even less within our 20-35°N/60-105°E box) due to sparse sampling and presence of clouds. This sampling is too limited to derive a robust averaged CO profile and do subsequent statistically significant analysis. However, following the reviewer's comment, we have added in the supplementary material a figure showing a comparison between MLS profiles and the model, and an inherent discussion is added in the manuscript (see 348:352 in the revised manuscrpt).**

**The double peak is not detected in modelled CO conversely to aerosols, because the lower peak is only linked to aqueous phase aerosol microphysics and not expected for gaseous precursors. This is very reasonable: we attribute the higher-altitude aerosol peak to gas phase chemistry (homogeneous nucleation) and this is reflected by the increased gaseous precursors concentration due to AMA-related convection.**

22) P10/L322/ L430 (new line number): '(Randel and Park, 2006; Garny...)'-> '(e.g. Randel and Park, 2006;Garny...)'
**Done**

23) P12/L359/ L468 (new line number): 'The vertical structure of the AMA-related dynamics has been investigated by several authors (Bergman, J. et al., 2013; Garny and Randel, 2013; Brunamonti et al., 2018)..' Remove 'J' after Bergman and add 'e.g.' . There are more previous publications studying the vertical structure of the AMA (e.g. Park et al. 2009; Vogel et al, 2019; Bian et al, 2020,..)
**Done and references added.**

Park M, Randel WJ and Emmons LK et al. Transport pathways of carbon monoxide in the Asian summer monsoon diagnosed from Model of Ozone and Related Tracers (MOZART). J Geophys Res 2009; 114: D08303.
Vogel, B., Müller, R., Günther, G., Spang, R., Hanumanthu, S., Li, D., Riese, M., and Stiller, G. P.: Lagrangian simulations of the transport of young air masses to the top of the Asian monsoon anticyclone and into the tropical pipe, Atmos. Chem. Phys.,19, 6007–6034, https://doi.org/10.5194/acp-19-6007-2019, https://www.atmos-chem-phys.net/19/6007/2019/, 2019.
Bian, J., Li, D., Bai, Z., Li, Q., Lyu, D., and Zhou, X.: Transport of Asian surface pollutants to the global stratosphere from the Tibetan Plateau region during the Asian summer monsoon, Natl. Sci. Rev., 7, 516–533, https://doi.org/10.1093/nsr/nwaa005, https://doi.org/10. 1093/nsr/nwaa005, 2020.
**Some of these references were added.**

24) P14/Fig.3: Please explain briefly in the Figure caption why an application of an extinction filter is shown.

**Done**

25) P18/L485/ L603 (new line number): Please explain the meaning of the p-value in words.

**The meaning was added in parentheses.**

26) P18/L506/ L616 (new line number): This mirrors the increase of the emissions in Asia.' Zheng et al. (2018) shows that after 2013 China's anthropogenic emission of some pollutants decreased substantially (e.g., SO2) because of the implementation of new emission control measures. How does that fit to your results about increasing emissions in Asia? Are the new Chinese emission control measures considered in the Regional Emission inventory that is used in this study?
Zheng, B., Tong, D., Li, M., Liu, F., Hong, C., Geng, G., Li, H., Li, X., Peng, L., Qi, J., Yan, L., Zhang, Y., Zhao, H., Zheng, Y., He, K., and Zhang, Q.: Trends in China's anthropogenic emissions since 2010 as the consequence of clean air actions, Atmos. Chem. Phys., 18, 14095–14 111, https://doi.org/10.5194/acp-18-14095-2018,
https://www.atmos-chem-phys.net/18/14095/2018/, 2018.

**Unfortunately, the CEDS emissions inventory does not include this recent regional emission inventory. The data for CMIP6 were published before our study and it usually takes time to introduce this kind of changes in regional inventories in the global emissions inventories. As has been detailed in the paper of Hoesly et al. (2018) and in the description of the Emissions (see Section 2 of our manuscript) REAS is the regional emission inventory used for the Asian region (covering the period 2000-2008) and MEIC (MEIC-Multi-resolution Emission Inventory for China)(Li et al. 2017) for China (having years 2008, 2010 and 2012).**

*Li, M., Zhang, Q., Kurokawa, J.-I., Woo, J.-H., He, K., Lu, Z.,Ohara, T., Song, Y., Streets, D. G., Carmichael, G. R., Cheng,Y., Hong, C., Huo, H., Jiang, X., Kang, S., Liu, F., Su, H.,and Zheng, B.: MIX: a mosaic Asian anthropogenic emission inventory under the international collaboration framework ofthe MICS-Asia and HTAP, Atmos. Chem. Phys., 17, 935–963,https://doi.org/10.5194/acp-17-935-2017, 2017*

*Hoesly, R. M., Smith, S. J., Feng, L., Klimont, Z., Janssens-Maenhout, G., Pitkanen, T.,  Seibert, J. J., Vu, L., Andres, R. J., Bolt, R. M., Bond, T. C., Dawidowski, L., Kholod, N., Kurokawa, J.-I., Li, M., Liu, L., Lu, Z., Moura, M. C. P., O'Rourke, P. R. and Zhang, Q.: Historical (1750-2014) anthropogenic emissions of reactive gases and aerosols from the Community Emissions Data System (CEDS), Geosci. Model Dev., 11(1), 369–408, doi:10.5194/gmd-11-369-2018, 2018.*

27) p17/Fig.4: Figure 4 shows very nicely the impact of volcanic eruptions. Is there also a modulation by El Niño?

**This is an interesting and complex question raised by the reviewer. ENSO affects remote regions of the globe with regional responses in atmospheric dynamics, precipitation, temperature, etc. The way it might impact transport and atmospheric burdens of aerosols and their precursors is an open question.**
**Several studies have shown that ENSO clearly impacts tropopause temperatures which control the amounts of water vapour in the UTLS. On a basic way of thinking, this could affect the oxidation capacity (through OH radical production) and the microphysics of UTLS aerosols. During El Niño positive anomalies of up to 10% in lower stratospheric H2O can be induced (Diallo et al., 2018). Such an investigation would require to analyse in details the alignment of ENSO with the phase of the QBO because the two mechanisms give rise to different patterns of variability in the tropical cold point tropopause temperatures with as a consequence different degrees of moistening or drying of the lower stratosphere depending on the QBO phase (Diallo et al., 2018). The QBO alone produces more H2O (and ozone) anomalies than the ENSO alone so the question could be raised for QBO also.**

**Perhaps one first step to address the questions of modulation of AOD by ENSO, QBO, volcanoes over the period covered in our work would be to use multilinear regression through a dedicated study as done in the Diallo et al.'s paper for Age of Air. However, the fact that ENSO exerts its impacts on remote regions of the globe through nonlinear atmospheric teleconnections and that patterns of these teleconnection have changed throughout time (possibly due to anthropogenic forcing) may complicate a robust statistical analysis with this kind of method.**

**This is definitely matter of a different dedicated paper.**

*Diallo, Mohamadou, Martin Riese, Thomas Birner, Paul Konopka, Rolf Müller, Michaela I. Hegglin, Michelle L. Santee, Mark Baldwin, Bernard Legras, and Felix Ploeger Response of stratospheric water vapor and ozone to the unusual timing of El Niño and the QBO disruption in 2015–2016, Atmos. Chem. Phys., 18, 13055–13073, 2018, https://doi.org/10.5194/acp-18-13055-2018.*

28) P18/L511/ L621 (new line number): Please clarify the meaning of 'increment' and 'correlation'.

**The sentence was changed accordingly.**

29) P21/L559/ L670 (new line number): Please clarify 'Our double-peak ATAL features highlighted in Fig. 6a'. I assume the meaning is that two maxima of AOD at different longitudes are found (corresponding to the bimodality of the AMA). It is confusing here because the expression 'double-peak structure' was already used for the two maxima

found in the vertical structure of the ATAL. Or is there an misunderstanding? Please clarify.

**The reviewer is right, the paragraph was confusing. We refer to the shape and maximum found in Fig 6a which are comparable with those found by Yu et al 2015. The paragraph was changed accordingly.**

30) P21/L564/ L673 (new line number): 'The difference between the AOD values obtained for the two altitude ranges in Fig 6a and 6b points at the importance of what we have identified as convective incloud aerosols.' Please explain this in more detail.

**We refer to the fact that the AOD difference found between the two range of altitude: 200-80 hPa (Fig 6a) and 120-80 hPa (Fig 6b) highlights the contribution of convective in-cloud aerosols, which makes that the AOD values for 200-80 hPa are larger than for 120-80 hPa.**
**The paragraph was extended accordingly.**

31) P21/L567: 'full double-peak ATAL' (see above L559)
**Changed, see answer 29.**

32) P22/L585/ L706 (new line number): 'The model evaluation with MLS and ACE-FTS satellite data reflects that transport and convection features are well represented in our simulations, despite a possible underestimation of the biomass burning emissions.' In the paper, a rough comparison between simulated CO and measured CO is shown. I would not call this 'model evaluation'. Further, I am not sure if the transport and convection features are overall well represented in the model (see comment to L256). Please rephrase this sentence and use a somewhat more cautious formulation.

**This paragraph and the title of section 3 have been changed accordingly, see answer n°18 and 21. We have also rewriting the conslusion as follow:**
**"The model results show overall good agreement with the space-time behaviour of CO in the UTLS region observed by the MLS and ACE-FTS space-borne instruments, despite a possible underestimation in the CO burden due to the underestimation of surface emissions. In particular, the horizontal distribution of modelled CO is in good agreement with MLS data and the vertical structure in the AMA shows a maximum near 150 hPa in agreement with the available ACE-FTS observations."**

33) P22/L590 /L716 (new line number): '..what has been reported in the past'. Please add some references.
**Done**

34) P22/L595: 'Apart from dust, the average partitioning for other aerosol types contained in the ATAL (from anthropogenic and from biomass burning emissions) is the following: 40% Sulfate, 30% Secondary Organic Aerosols, 15% Primary Organic Matter, 14% Ammonium and less than 3% Black

Carbon.' What is new or different compared to previous results regarding the chemical composition of ATAL?

**As for a previous reply (see answer n° 2) new contribution of our work is the ATAL aerosol modeled trends over 16 years (not studied before) and the new ATAL double-peak structure, which, although it has been observed in some recent measurements has not been further discussed.**
**Regarding specifically to the chemical composition, we have compared our results with some previous works (e.g. Yu et al 2015). With respect to our work, they have also reported approximately the same % of sulfate, but a larger % of organic aerosols (45% in our model, 60% in theirs). Yu et al. 2015 have also simulated large amounts of dust but they don't explicitly report the percentage. Unlikely Fadnavis et al. 2013,2017, our model doesn't show a maximum of black carbon in the ATAL. Some others works, like Fadnavis et al 2013 and Fairlie et al 2020, include nitrate in his models, while CESM-MAM7 doesn't treat nitrates.**

35) P22/L602/ L731 (new line number): '... a marked positive trend of anthropogenic and biomass burning aerosol concentrations is found, with up to a factor two increase of mass concentrations between 2000 and 2015. ' What are the consequences if the ATAL over Asia is increasing further in future?

**The consequences of the continuous anthropogenic emissions increase in Asia (principally of $SO_2$ and volatile organic compounds), and likely therefore aerosols in the ATAL, could be an impact in the radiative balance, stratospheric ozone chemistry, and properties/occurrence of cirrus clouds. However, as discussed before (see answer 27), the new emission control measures for $SO_2$ emissions in China is not considered in our CEDS emissions inventories and therefore could have different implications in the trends showed.**

---

## Author Comment (AC3) · 12 Nov 2020

We would like to thank to the reviewer for her/his detailed and mostly positive comments and suggestions. We discussed each of the points raised by Reviewer 3 among the coauthors and made the changes in the text accordingly. Below each comments, please find our answers and the respective changes made.

**Note:**

Due to changes made in the manuscript, some of the line numbers referred by the reviewers have changed. These changes are shown in green when applicable.

**Received and published: 14 September 2020**

General comments:

This study is very well written and addresses a hot topic in the scope of ACP. It provides interesting hypotheses about the nature of the ATAL, e.g. that there is a double-peak vertical structure, mineral dust dominates aerosol mass, and that the ATAL signature has been increasing from 2000 to 2015. This should be published, considering the following.

The paper would benefit from actually working out if one or more of the above hypotheses have something to do with reality. In the current version, the analyses and discussions are limited almost exclusively to the modelling world, to one simulation.

This simulation is linked to the real world just by a comparison to observed CO. However, the emissions contributing to ATAL have other source distributions than CO, and are affected by other processes.

Furthermore, the uplift of air from the ground to the UTLS - a crucial process for ATAL – might need a closer look in the model: simulated CO compares favorably to the observations in the UTLS, despite being off in the free troposphere (Figure 1e). A much more thorough model evaluation would be appropriate, covering (proxies for) all species, precursors and processes of relevance for the aspects of ATAL that are discussed in this study. Sensibly splitting this between the supplement and the main text would allow the paper to stay concise.

Apart from the mere model evaluation, it would help putting some effort into researching available observations for support of the model-based findings about ATAL.

A more detailed understanding of the strengths and weaknesses of the simulation might also help the discussion of how this study compares to other modelling results.

**We thank the reviewer for constructive comments. As discussed in the manuscript and in some previous studies, the chemical**

composition of the ATAL remains poorly characterized due to the lack of in in situ measurements in the AMA region. Only over the last recent years some aircraft and balloon campaigns have started to be conducted in the region (e.g. Stratoclim from 2016 to 2018 and BATAL from 2015 to the present), i.e. mainly after the period of our simulation. That is why the present study cannot be exhaustively compared with in situ measurements. In addition, satellite observations of aerosols in terms of their composition are very scarce and mostly limited, for this application, by the interaction of radiation with co-existing clouds (so that the necessary cloud screening likely screened out the lower of the two ATAL peaks, in past works, see discussion about the results of Vernier at al., 2015). However, the double-peak structure of the ATAL was observed before, even if not discussed in past works. So, we have extended our discussion regarding the double-peak vertical profile doing a qualitative comparison with some of these recent measurements (Vernier et al., 2018, Brunamonti et al., 2018, Höpfner et al., 2019) see Sec 4.2, 489:503 in the revised manuscript.

On the other hand, as many studies that have been carried out (Fadnavis et al. 2013,2017; Yu et al., 2015, 2017; Gu et al. 2016; Lau et al. 2018; Ma et al., 2019; Fairlie et al., 2020; etc, cited in the current work), different models simulations provide new insights into the composition, budget, origin and source contribution to the ATAL.

As for the validation using satellite observations of gas specie, please note that these comparisons, for CO, were only meant to illustrate the ability of the model to transport pollutants to the UTLS. CO has been used as a representative pollution tracer in the UTLS. Simulated CO shows a broad maximum over the monsoon anticyclone region, in a reasonable agreement with the spaceborne observations in term of spatial extent.

In this work we do not compare all the gas species with satellite data and do not evaluate all chemical and physical processes computed by CAM5 since this would require a large work and a specific dedication and efforts in itself. As similar past studies (where model validation was generally absent or, in any case, less detailed than ours), here we focus on ATAL aerosols distribution, composition and, more originally, long-term variability of each aerosol type and their integrated optical properties, which has never been reported before. We discuss throughout the text some limitations of the model whenever possible.

Specific comments: 1) L87/ L105 (new line number): have -> has Changed

2) L105/ L130 (new line number): The bimodality of the AMA has been discussed for longer, see e.g. (Nützel et al. 2016, Pan et al. 2016) and references therein.

The references have been changed and this previous works added.

3) L107/ L130 (new line number): beweekly -> biweekly or bi-weekly **Done**

4) L116 /L140 (new line number): larger aerosols composition -> more comprehensive aerosol composition
Changed it

5) L116/ L140 (new line number): Please check the use of aerosols / aerosol / aerosols' / . . . throughout the paper. **Done**

6) L149 /L187 (new line number): Isn't anvil associated with convective rather than stratiform clouds?
We have removed the term "anvil" which was, indeed, confusing.

7) L179 /L218 (new line number): Simulated ATAL trends are likely to critically depend these assumptions. Please elaborate on the uncertainties in the emissions' setup, providing the reader with a sense on how this might print through to the results for ATAL.

The paragraph was changed accordingly, and we have added at the end of this paragraph (L225-227) a sentence to explain more in detail the assumptions made by CEDS inventory that introduce uncertainties. This is mentioned in the conclusion as well.

8) L191 /L239 (new line number): Different reanalyses have different peculiarities in representing AMA (see e.g. Nützel et al. 2016). Please shortly note whether there is something specific the reader needs to know about MERRA.

To our knowledge, only the old NCP reanalyses are problematic in representing the AMA (strong bimodality). All modern reanalyses, including MERRA2, agree well on the AMA. There are guite large differences, however between modern reanalysis regarding cloud properties and heating rates. But this is probably not relevant here. MERRA2 reanalyses are compared with other datasets in Long et al. (2017), where they report a very good agreement in temperature seasonally and latitudinally between the surface and 10 hPa, for the more recent reanalyses (CFSR, MERRA, ERA-Interim, JRA-55, and MERRA-2). Zonal winds are in greater agreement than temperatures and this agreement extends to lower pressures than the temperatures. Older reanalyses (NCEP/NCAR. NCEP/DOE,ERA-40, JRA-25) have larger temperature and zonal wind disagreement from the more recent reanalyses.

In Sec 2.1 we have added a sentence to explain that our model is driven by MERRA2 data (and not MERRA like show Nützel et al. 2016) with a constrain of 10%, i.e every time step the offline meteorological fields (horizontal wind components, air temperature, surface temperature, surface pressure, sensible and latent heat flux, and wind stress) are nudged to the online calculated meteorology. The nudging coefficient in our case is 0.01 (10%). Long, Craig S., Masatomo Fujiwara, Sean Davis, Daniel M. Mitchell, and Corwin J. Wright, Climatology and interannual variability of dynamic variables in multiple reanalyses evaluated by the SPARC Reanalysis Intercomparison Project (S-RIP), Atmos. Chem. Phys., 17, 14593-14629, 2017, https://doi.org/10.5194/acp-17-14593-2017

9) L281: A 30 Corrected

10) L303 /L379 (new line number): Please consider showing the comparison to the corresponding observations.

There aren't "corresponding observations" for these aerosols during this year. Is not possible to shows such direct comparison. As answered at the beginning, due the lack of information about the ATAL composition derived from observations in the AMA region over the period of simulation we can only qualitatively compare our model results with the ion chromatography analysis from aerosol samples collected in summer 2015 in India during the BATAL balloon campaign (Vernier et al., 2018).

11) L330/ L440 (new line number): Is the following understanding correct? There is no dynamic tracking of the AMA. Rather you choose a static box, which most of the time is part of the AMA. Any averages should thus be dominated by AMA conditions. This is ok, but some rewording might help to make the approach clearer.

**We have made some rewording to clarify the approach which is indeed based on a static box and not on dynamic tracking changing with time. See Lines 431-444 in the revised manuscript.**

12) L335/ L449 (new line number): Isentropic surfaces might be better to describe horizontal transport and thus the horizontal extent of the ATAL (Santee et al. 2017, Gottschaldt et al. 2018). Please check whether or not your results crucially depend on the choice of the vertical coordinate system.

We have carried out the same analysis doing the plots at different isentropic surfaces (400, 380, 360 K) and the plots look pretty similar. So, we have decided to carry out our analysis in pressure levels since this is the basic coordinate system in our model and this does not require any interpolation at each time step.

13) L363/L476 (new line number): The term "mode" is already in use for aerosol size ranges and for the dynamics of the AMA. Does it refer to different aerosol classes here?

The paragraph was probably confusing, so we have deleted the term bi-modal. The two relative maximum observed (double-peak) refers to two different origin of aerosols that are present at different altitudes one at lower altitudes (~ 250 hPa) associated with "convective" cloud-borne aerosols and one at higher altitudes

(~ 100 hPa) associated with "clear-sky" aerosols. This is discussed later in the paragraph.

14) L363/ L476 (new line number): Is there any observational hint for such a double-peak layering?

Yes (even if not for the time period of our simulations). The works of Vernier et al. 2018, Brunamonti et al. 2018, Höpfner et al 2019 shows evidence of this double peak and a discussion about this was included later in the Sec 4.2, see lines 489-503 in the manuscript.

15) L397 /L525 (new line number): "Double-peak", when used as adjective?Please check throughout the paper.Done

16) L405/L531 (new line number): Please mention in the caption that this is modelling only.Ok, Done.

17) L433 /L552 (new line number): That is rather vague. Several models get an ATAL, so it seems to be a quite stable feature. Interestingly, the exact characteristics vary, probably depending on the various factors listed here. For improving our understanding of ATAL it is therefore important to really understand the model differences, and to find those explanations that are supported by observations.

We agree with the reviewer's comment but as replied to a previous comment (answer 10), there are only few available in situ observations about the chemical composition of aerosols present in the ATAL. We would need a significant number of new in situ observations to make a comprehensive comparison with our model outputs, as well as gathering more information on the "real" ATAL (i.e. from observations). For the moment, we still can have some information using models: we think it useful to provide information on the ATAL's composition and temporal variability by models.

For the other hand we have reorder our discussion in Sec 4.1 regarding the comparison with other models results and regarding specifically to dust an extended discussion and the possible limitations of the model have been added in Sec 4.1 (lines 389-419) and in the conclusions.

18) L457/L558 (new line number): Please use subscripts in chemical compounds throughout the paper.Done

19) L484 /L606 (new line number): showed -> shown **Done**

20) L493 /L612 (new line number): Is there a chance to be more specific: Which aspects of the dust cycle are captured well by your model? Which are not and what are the implications for your conclusions about ATAL?

As mentioned before (answer 17) and in the replies to Reviewer#1 (answer 5 and 9), there are still large uncertainties in dust

**modelling across different models. However we have extended our discussion about the possible biases for the modelled dust (Sec 4.1 and Conclusion in the revised Manuscript).**

21) L503 /L616, 686 (new line number): Here you state model shows that increased emissions translate into enhanced ATAL, but in L567 alternative explanations are offered. Please check consistency. Furthermore, as already noted by reviewer2, emission trends are more complicated. A more detailed analysis might be needed, e.g. explicitly correlate emissions (by region) with ATAL parameters.

We wish to clarify that we actually do not provide a different explanation since in the first part we analyse the contribution of aerosols in the accumulation mode (a1), which are principally from anthropogenic contribution. For the discussion about AOD, we account for the total modelled extinction which includes all the aerosol types present in CESM-MAM7 and in any case we find an increase of a factor of 2 for the AOD for both ranges of altitude. In the manuscript, we specify that the reasons for the the ATAL AOD increases (increase in Asian emissions, more efficient vertical transport or other reasons) require further investigation. Following the reviewer's suggestion, sensitivity studies could be done by masking emissions from mainly contributing regions (e.g. China for  $SO_2$ , Gangetic valley for  $NH_3$ ) or testing different emission inventories which could be the scope of a dedicated study. In the conclusion a paragraph regarding the implication of the CEDS emissions used has been added (L733-738).

22) L552 /L663 (new line number): This formulation is kind of suggesting that Vernier et al. might be wrong. Please elaborate. **The paragraph was changed it accordingly.**

23) L555 /L669 (new line number): Is CESM1/CARMA from the same model family you are using? Then getting similar results could also indicate a common problem.

CESM1/CARMA is indeed the same family model than our CESM1/CAM5-MAM7. Nevertheless, the aerosol models (CARMA and MAM7) are deeply different. We obtain similar results to Yu et al. 2015 like the features of the maximum in the AOD vs longitude and the AOD values. These simulated values are higher than those reported by Vernier et al. (2015) using the CALIOP space-borne lidar. However, different cloud-screening procedures have been used in Vernier et al. (2015) and in our study. On may argue that aerosols with high extinction, like those we have identified from convective cloud-borne aerosols in our lower altitude peak, might have been removed from the lidar signal during the cloud-screening process, in the paper by Vernier et al (2015).

While the residual differences with respect to Yu et al. 2015 are easily attributable to the different aerosol models, they may also be due to the different emission inventories used. Yu et al., 2015 have used GFED3 for biomass burning emission and GFED2 for SO2 biomass burning emissions, anthropogenic emissions are taken from EDGAR-FT2000 and biogenic emissions are estimated by Guetner et al 2006. Different meteorological data used (MERRA, used by Yu et al 2015, MERRA2, used in the present study) may also have played a role.

24) L562/ L672 (new line number): Another interesting hypothesis. Please check whether there are any observations supporting it.

As was mentioned at the beginning and in the answer 14, this hypothesis of different AOD values obtained for the two altitudes ranges can be attributed to the different aerosols present at different altitude ranges, related to the double-peak vertical profile of aerosols found. We have added a discussion regarding the observation of this double-peak structure observed in some recent balloon and aircraft campaigns (Sec. 4.2).

25) L585/ L712 (new line number): Please consider rewording: The results show . . . -> Our modelling results indicate . . . **Done**

26) L594: Please make it clear from the beginning that nitrate aerosols might be an important aspect you omit.Yes, this was added in the abstract

27) L812/ L970 (new line number): space between references missing **Corrected**

---

## Author Response (AR2)

We thanks to the reviewer for the time taken for read the new revised Manuscript. In the following, we address the last comments of the reviewer and the changes made to the manuscript accordingly.

L332 reads: "The vertical distribution of CESM-MAM7 simulations shows a quite remarkable agreement with ACE-FTS observations above 400 hPa." Does this refer to Fig. 1e? If so, in my opinion agreement starts at altitudes above 150hPa. At 400hPa the simulation considerably differs from the observations, which could indicate an issue with the parameterization of convective transport (detrainment?; convective altitudes?). Furthermore, Fig. 1e is for one snapshot in time only. I recommend to discuss simulated vertical profiles in comparison to ACE-FTS based on temporal averages, or -better- a statistically meaningful number of snapshots like Fig. 1e. I find Fig. S1 much more of a meaningful comparison, but it is hidden in the supplement and MLS lacks resolution.

Yes, we refer to the comparison of the vertical distribution in Fig 1e. The reviewer is right, the model and ACE-FTS observations are in agreement only above the 200 hPa level. As a result, so we have changed "above 400 hPa" to "above 200 hPa" in the revised Manuscript.

The reasons for the differences between observed and modelled CO with respect to altitude are still unclear and we have discussed this issue in the next paragraph in the manuscript "The discrepancies observed between simulated and observed CO could be linked to the treatment of convection by CESM1/CAM5 together with discrepancies in emission inventories (see discussion above)."

The convection in CESM-MAM7 is parameterized using the Zhang-McFarlane scheme (Zhang and McFarlane, 1995) for deep convection and the Hack scheme (Hack, 1994) for shallow convection (See L243:249 in the revised manuscript). This is a typical parameterization used in numerous studies involving the CAM5 (or previous versions) model.

As answered to reviewer 2 and accordingly modified in the revised manuscript, only a few ACE-FTS profiles are available each year in the AMA (and even less within our 20-35°N/60-105°E box) due to sparse sampling and presence of clouds. This sampling is too limited to derive a robust averaged CO profile and to do subsequent statistically significant analysis. For this reason we have

added in the supplementary material a figure showing a comparison between MLS profiles and the model, and the corresponding discussion (see L:348:352).

We prefer to leave this comparison with MLS in the Supplemental material. Firstly, due to the coarse vertical resolution of MLS which limits the interpretation in term of convective (detrainment altitude, top of cloud level) and secondly because the main objective of the paper is to show the results of our simulations in terms of composition and trend of aerosols in ATAL.

L339: under predictions -> underestimations Changed

L341: over predicted -> overestimated Changed

L747: Given the rather loose connection between observational and simulated world in this study, the authors might consider a more cautious formulation here. Something like:

"A double-peak vertical structure ... so far. These observations support our simulation results, which in turn provide a possible explanation for the observations. Given the uncertainties discussed throughout the paper, it is not entirely clear whether the simulation is right for the right reasons, but it provides hypotheses for follow-up studies."

We thanks for this comment and we agree with this formulation recommend so we have modified it in the manuscript as follow:

"This "double-peak" vertical structure has been observed in recent balloon and aircraft campaigns (e.g. Vernier et al., 2018; Höpfner et al., 2019) but has not been discussed in detail so far. These observations support our simulation results, which in turn provide a possible explanation for the observations. Given the uncertainties discussed throughout the paper, the ability of our simulations to represent the reality of the convective transport in the ASM is not entirely clear but the model results provide hypotheses for follow-up studies."

---

## Author Response (AR3)

Dear Dr. Stiller

Thank you very much for your revision.
Regarding your question we would like to say that we prefer to not use any interpolation (neither linear nor logarithmic) to reflect the limited vertical information that MLS provides. However, you are right that the plotting procedure needs to be stated more clearly. We updated the Figure S1 respectively and added a description on the plotted pressure levels in the caption.

Best Regards

Adriana Bossolasco